# SCULPTOR: EMPOWERING LLMS WITH COGNITIVE AGENCY VIA ACTIVE CONTEXT MANAGEMENT

**Mo Li**[1,2,3]**, L.H. Xu**[4]**, Qitai Tan**[2]**, Long Ma**[5]**, Flood Sung**[4]**, Ting Cao**[1]***Yunxin Liu**[1,3]
[1] Institute for AI Industry Research (AIR), Tsinghua University
[2] Shenzhen International Graduate School, Tsinghua University
[3] Shanghai AI Laboratory  [4] Independent Researcher  [5] Peking University

## ABSTRACT

Large Language Models (LLMs) suffer from significant performance degradation when processing long contexts due to proactive interference, where irrelevant information in earlier parts of the context disrupts reasoning and memory recall. While most research focuses on external memory systems to augment LLMs' capabilities, we propose a complementary approach: empowering LLMs with Active Context Management (ACM) tools to actively sculpt their internal working memory. We introduce **Sculptor**, a framework that equips LLMs with three categories of tools: (1) context fragmentation, (2) summary, hide, and restore, and (3) precise search. Our approach enables LLMs to proactively manage their attention and working memory, analogous to how humans selectively focus on relevant information while filtering out distractions. Experimental evaluation on diverse long-context benchmarks demonstrates that **Sculptor** significantly improves performance even without specific training, leveraging LLMs' inherent tool-calling and instruction-following capabilities. To further optimize these strategies, we introduce a novel dynamic context-aware reinforcement learning (RL) approach, advancing the training of an agent that actively modifies its own conversational history. By enabling Active Context Management, **Sculptor** not only mitigates proactive interference but also provides a cognitive foundation for more reliable reasoning across diverse long-context tasks—highlighting that explicit context-control strategies, rather than merely larger token windows, are key to robustness at scale.

## 1 INTRODUCTION

Large Language Models (LLMs) have demonstrated remarkable capabilities across diverse tasks, yet they face fundamental challenges when processing long contexts. Prior work shows that simply enlarging the context window leaves models vulnerable to position bias, overload, and interference as sequences grow (Liu et al., 2023a; Hsieh et al., 2024a). Recent studies (Wang & Sun, 2025) have empirically demonstrated that LLMs suffer from proactive interference, where earlier information in the context disrupts the processing of subsequent, more relevant information. Moreover, calibrations like Found in the Middle (Hsieh et al., 2024b) reduce—but do not eliminate—positional bias; recent evaluations (Tian et al., 2025) find that performance still degrades significantly when the distance between relevant information pieces increases, as irrelevant information between them interferes with effective information integration. These phenomena mirror human cognitive psychology, where new learning can be impaired by previously acquired information that is no longer relevant to the current task.

The challenge becomes particularly acute in complex, multi-step reasoning tasks where LLMs must maintain focus on multiple critical information pieces while filtering out contextual noise (Li et al., 2025a). Traditional approaches to address long-context challenges have primarily focused on expanding context windows or developing external memory systems (Li et al., 2025c; Yang et al., 2024; Wang & Chen, 2025; Chhikara et al., 2025; Packer et al., 2024; Wang et al., 2024; Suzgun et al., 2025). While these solutions increase the amount of information an LLM can access, they do

---

*Corresponding author.

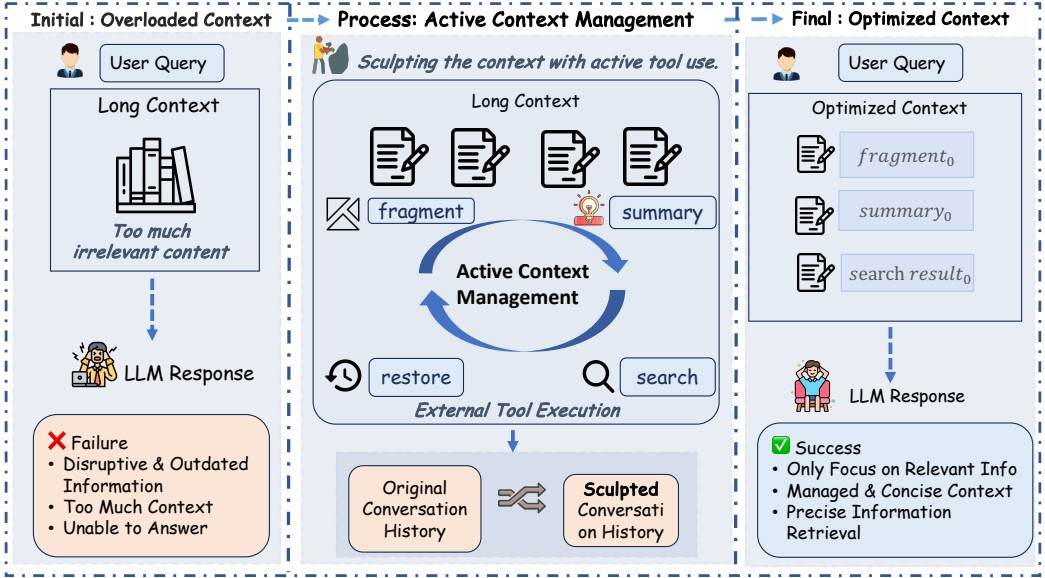

Figure 1: Overview of **Sculptor** framework: Through Active Context Management, LLMs transform overloaded contexts into optimized contexts using fragment, summary, search, and restore operations, enabling successful task completion where traditional approaches fail due to interference.

not address the fundamental issue of proactive interference—**the inability to actively manage and curate the working memory that directly influences reasoning processes**.

Consider a human expert working on a complex problem: they naturally employ active memory management strategies, selectively attending to relevant information, summarizing key insights, and temporarily setting aside less important details. They can revisit previously discarded information when needed, but crucially, they do not allow irrelevant details to continuously interfere with their current reasoning process. Current LLMs lack this fundamental cognitive capability. We propose that the solution lies not merely in expanding the context window, but in **empowering LLMs with the ability to actively manage their internal working memory**. Following established distinctions in Li et al. (2024a); Guo et al. (2024), we focus on optimizing the model's working memory—the immediate context where attention operates and reasoning occurs—rather than external memory systems that store information outside the model's immediate context.

To this end, we introduce **Sculptor**, a novel framework that treats LLMs as active sculptors of their own context. Just as a sculptor views a block of marble and selectively removes material to reveal the desired form, **Sculptor** achieves this through a process we call Active Context Management (ACM), as illustrated in Figure 1. We equip LLMs with the **Sculptor** tool suite that enables them to: (1) **Fragment and Organize**: Segment long conversations into manageable pieces with unique IDs for easy reference. (2)**Summary, Hide, and Restore**: Generate focused summaries, dynamically fold irrelevant sections to reduce clutter, and flexibly restore or expand content as needed. (3) **Search and Retrieve**: Perform exact searches to quickly locate relevant information.

This approach represents a paradigm shift from passively processing ever-growing contexts to active context curation. Instead of being overwhelmed by increasingly long contexts, LLMs learn to proactively manage their attention and working memory, focusing computational resources on the most relevant information. We view **Sculptor** as a representative of this emerging direction—complementary to external memory systems (Li et al., 2025c; Yang et al., 2024; Wang & Chen, 2025; Chhikara et al., 2025; Packer et al., 2024; Wang et al., 2024; Yu et al., 2025) that focus on cross-session persistence and context extension approaches—providing a necessary step toward reliable long-horizon reasoning. Related work on context compression (Xu et al., 2023; Jiang et al., 2024b; Guo et al., 2025) further demonstrates that selectively foregrounding key information can simultaneously improve accuracy and reduce cost and latency, reinforcing the need for explicit context control over passive attention alone. Recent work also suggests that in-context learning can be viewed as implicit weight up-

dates (Dherin et al., 2025), implying that allowing models to modify their own context enables a form of "self-evolution" (Zhang et al., 2025a)—a step toward agents that can adapt their computational substrate without external intervention.

Our key contributions are as follows:

- We propose Active Context Management (ACM) for LLMs and realize it with Sculptor. Unlike external memory systems designed for cross-session persistence or retrieval methods that irreversibly filter information, Sculptor enables principled, fully reversible optimization of single-session internal working memory.

- We propose an RL training approach for active context modification, introducing Conditional Trajectory Collection and Incremental Loss Assignment to enable effective learning of context manipulation strategies. Through dynamic context-aware GSPO training, we achieve substantial performance gains across diverse long-context benchmarks.

- We provide comprehensive analysis of tool usage patterns, attention mechanisms, and cost analysis, demonstrating that ACM effectively reduces context token consumption while enhancing long-context capabilities.

## 2  METHODOLOGY

**Sculptor** introduces a paradigm shift in how LLMs handle their working memory. Instead of passively accepting all information in their context window, we empower models to actively manage their attention through a suite of context manipulation tools. Our framework operates on the principle that intelligent information curation is as important as information capacity.

### 2.1  TOOL DESIGN PRINCIPLES

Our tool design follows four core principles. (1) **Deterministic and Self-Contained Operations**: each tool is a simple, deterministic operator without external dependencies (e.g., embedding models), a self-contained design that guarantees deployment stability and isolates the LLM's cognitive agency for pure evaluation. (2) **Cognitive Alignment**: the tools mirror effective human strategies, such as our `search_context` tool performing exact matching akin to "Ctrl+F", a computationally efficient approach that reserves complex semantic understanding for the LLM's own reasoning. (3) **Structural Preservation for Scalable Training**: the tools are constrained to never alter the count or order of messages, thereby maintaining a stable state representation that is critical for tractable credit assignment in reinforcement learning. (4) **Reversibility and Graceful Degradation**: All context-modifying operations are designed to be non-destructive and fully reversible (e.g., fold is undone by expand), ensuring no information is permanently lost. This guarantees that the framework functions as a strict superset of the baseline model's capabilities, allowing for graceful degradation: if no tools are invoked, the model's behavior is identical to its original, unmodified state.

### 2.2  THE **SCULPTOR** TOOL SUITE

Following these design principles, we equip LLMs with six fundamental tools organized into three functional categories. These tools work in coordination within a single turn: the agent receives a user message and performs multi-step tool calls, continuously invoking tools until generating a final response. For instance, fragmenting a context segment yields a unique fragment ID that enables subsequent operations like compression, summarization, or restoration. Complete JSON schemas for all tools are provided in Appendix H.

*(1) Context Fragmentation* is handled by `fragment_context`, which segments long conversations into manageable fragments using start and end markers, with each fragment receiving a unique 6-character ID for easy reference.

*(2) Context Compression and Restoration* involves three complementary tools for dynamic content management. `summarize_fragment` generates focused AI-powered summaries of specific fragments based on user-specified focus areas (e.g., technical details, key decisions, action items), compressing content while preserving critical information. `fold_fragment` temporarily hides fragment content while preserving its existence, displaying only a folded marker to dramatically

reduce visual clutter. `restore_fragment` provides universal restoration capability, reverting both summarized and folded fragments back to their original content, ensuring no information is permanently lost during context management operations.

*(3) Precise Search and Retrieval* is accomplished through two complementary tools. `search_context` performs exact keyword matching across user messages, assistant responses, or all content—mirroring the human approach of using Ctrl+F for information retrieval. It returns up to 50 matches with configurable result context windows. `get_search_detail` retrieves extended context around specific search results, with the model specifying the desired surrounding character count. By appending search results to the end of conversation history, this approach mitigates the "lost in the middle" problem (Liu et al., 2023a) where models struggle to locate information buried within long contexts.

## 3 TEACHING LLMS TO USE SCULPTOR TOOLS

Building on the strong tool-use capabilities inherent in modern LLMs, we explore two distinct approaches for teaching models to effectively wield the **Sculptor** tool suite. Throughout this paper, we use "ACM tools" to specifically refer to our **Sculptor** implementation—a concrete instantiation of the broader Active Context Management paradigm.

### 3.1 INHERENT TOOL-USE PERFORMANCE

We first evaluate the inherent tool-calling capabilities of state-of-the-art models like Claude-4-Sonnet and GPT-4.1, which demonstrate strong zero-shot generalization abilities for function calling. These models can understand and execute our **Sculptor** tools without any specific training, relying on their pre-trained understanding of tool usage patterns and natural language descriptions of tool schema. This zero-shot approach requires no additional training—models directly interpret and use the tools based solely on their schemas. To encourage consistent tool engagement, we set `tool_choice="required"` for the first step of multi-step conversations.

### 3.2 MULTI-STEP AGENT RL TRAINING WITH DYNAMIC CONTEXT-AWARE GSPO

To optimize tool usage strategies beyond zero-shot generalization, we develop a reinforcement learning approach specifically designed for multi-step tool calling in long-context scenarios. Our approach addresses the unique challenges of training models to actively manage dynamic contexts where tool calls can fundamentally alter the information landscape.

**Group Sequence Policy Optimization (GSPO).** We adapt GSPO (Zheng et al., 2025) for multi-step rl training, leveraging its sequence-level optimization for stable training in long-context scenarios. Given a query $x$ and $G$ sampled trajectories $\{\tau_i\}_{i=1}^G$ from policy $\pi_{\theta_{\text{old}}}$, GSPO optimizes:

$$\mathcal{J}_{\text{GSPO}}(\theta) = \mathbb{E}_{x \sim \mathcal{D}} \left[ \frac{1}{G} \sum_{i=1}^G \min \left( s_i(\theta) \hat{A}_i, \text{clip}(s_i(\theta), 1 - \varepsilon, 1 + \varepsilon) \hat{A}_i \right) \right] \tag{1}$$

where the sequence-level importance ratio is:

$$s_i(\theta) = \left( \frac{\pi_\theta(\tau_i | x)}{\pi_{\theta_{\text{old}}}(\tau_i | x)} \right)^{\frac{1}{|\tau_i|}} \tag{2}$$

and the group-normalized advantage is:

$$\hat{A}_i = \frac{r(x, \tau_i) - \text{mean}(\{r(x, \tau_j)\}_{j=1}^G)}{\text{std}(\{r(x, \tau_j)\}_{j=1}^G)} \tag{3}$$

**Dynamic Context-Aware Credit Assignment with Incremental Loss Design.** The key innovation in our approach addresses the non-monotonic nature of context evolution during tool calling. Traditional multi-step RL assumes each trajectory $\tau_t$ is a prefix of $\tau_{t+1}$, allowing training only on the final trajectory. However, with context management tools, $c_t \not\subset c_{t+1}$ in general—tools like

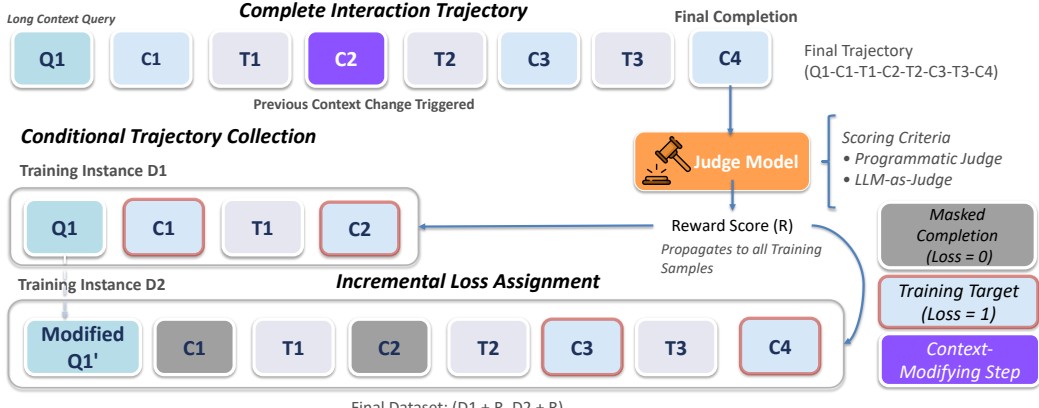

Figure 2: Conditional trajectory collection and incremental loss assignment for RL training. Q represents the initial user context, C denotes assistant completions, and T indicates tool results. Top: Complete interaction trajectory with context-modifying tool at step C2. Bottom: Training samples extracted via conditional trajectory collection, where each context change creates a new training instance. Incremental loss is assigned only to new completions (red boxes) while masking prior completions (loss=0), preventing redundant learning and training collapse.

`fold_fragment` or `summarize_fragment` actively remove or transform information, creating divergent context states.

To handle this, we introduce a two-part strategy combining **conditional trajectory collection** and **incremental loss assignment**, illustrated in Figure 2 and detailed in Appendix E. The final reward is propagated to all sub-trajectories within the same rollout, ensuring each context state receives appropriate learning signal. This incremental design prevents the model from learning spurious patterns where context-modifying tools repeatedly trigger themselves, which would cause training collapse. Each tool call receives gradient signal exactly once across all completions, ensuring stable and efficient learning. Notably, this method applies equally to both supervised fine-tuning (SFT) and reinforcement learning stages, providing a unified framework for training with dynamic contexts.

## 4 EXPERIMENTS

We evaluate **Sculptor** in two settings: zero-shot tool calling leveraging models' inherent capabilities, and after reinforcement learning with dynamic context-aware GSPO to optimize tool usage strategies.

### 4.1 EVALUATING PROMPT-GUIDED TOOL CALLING PERFORMANCE

**Evaluated Models:** We evaluate the effectiveness of **Sculptor** by comparing LLMs with and without the **Sculptor** tool suite across challenging benchmarks. Our experiments focus on Claude-4-Sonnet (Anthropic, 2025), GPT-4.1 (OpenAI, 2025), and DeepSeek-V3 (DeepSeek-AI et al., 2024) as representative state-of-the-art models, testing both baseline configurations and **Sculptor**-enhanced versions.

**Evaluated Benchmarks:** We evaluate on five benchmarks testing diverse long-context challenges: (1) **PI-LLM** (Wang & Sun, 2025) tests proactive interference through continuous key-value updates (2-256 updates, 46 keys). (2) **NeedleBench** (Li et al., 2025a) Multi-Needle Reasoning requires connecting 2-5 needles simultaneously across varying context lengths. For cost efficiency and rapid validation, we initially evaluate only on PI-LLM and NeedleBench in zero-shot settings. After RL training, we expand to: (3) **MRCR** (Vodrahalli et al., 2024a) for multi-round co-reference resolution, requiring models to distinguish between multiple identical requests (2-8 needles) and return the i-th occurrence from synthetic conversations. (4) **LongBenchV2** (Bai et al., 2025) for comprehensive long-context understanding. (5) **FRAMES** (Krishna et al., 2025) for factuality,

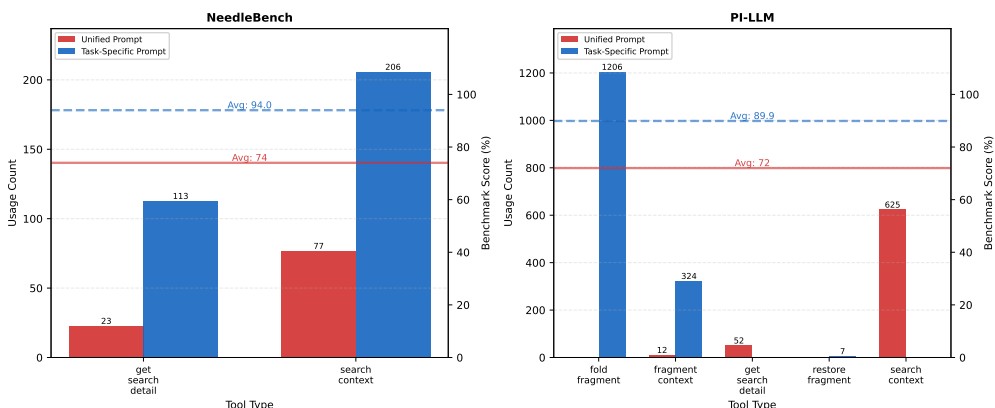

Figure 3: Tool usage count comparison for Claude-4-Sonnet before and after prompt optimization. Without task-specific prompts (unified prompt), both benchmarks show suboptimal patterns. With benchmark-specific prompt engineering, distinct improvements emerge: PI-LLM shifts from inefficient search-heavy patterns (625 calls) to strategic `fold_fragment` usage (1206 calls) for managing obsolete information, while NeedleBench increases search operations from 77 to 206 calls—addressing insufficient execution depth through more thorough verification and multi-hop reasoning. These contrasting patterns highlight how prompt engineering resolves different challenges: tool selection efficiency for PI-LLM versus execution completeness for NeedleBench.

retrieval, and reasoning measurement, containing 824 multi-hop questions requiring integration of information from 2-15 Wikipedia articles.

**Inherent Challenges of Unguided Tool Use:** To understand how models naturally interact with ACM tools, we conducted initial experiments using Claude-4-Sonnet on PI-LLM and NeedleBench benchmarks, collecting 50 samples from each benchmark for tool usage analysis. We provided the model with a unified system prompt—minimal, generic instructions applicable across all tasks—without any benchmark-specific guidance (see Appendix D.2 for the complete prompt).

Our findings revealed suboptimal tool selection patterns, as shown in Figure 3 (left bars). For PI-LLM, which contains numerous obsolete key-value pairs requiring the model to focus on the latest mappings, we expected the model to leverage `fragment_context` and `fold_fragment` to compress outdated information. However, Claude-4-Sonnet overwhelmingly relied on `search_context` (90.7% of tool calls), attempting exhaustive searches for each of the 46 keys despite hundreds of historical updates per key. This search-heavy approach proved highly inefficient—the model exhausted its 20-tool-call budget merely aggregating occurrences without effectively filtering obsolete information. Similarly, for NeedleBench, while search tools are appropriate for retrieval tasks, the model showed limited strategic diversity in tool selection.

These observations reveal three fundamental challenges in unguided tool calling: (1) **Suboptimal tool selection efficiency**: The model failed to recognize when certain tools become inefficient for specific scenarios. In PI-LLM, attempting exhaustive searching for 46 keys with hundreds of historical updates each consumed the entire tool budget, when structural reorganization through fragment-and-fold would have been far more efficient. (2) **Tool dependency misunderstanding**: The model lacked comprehension of tool prerequisites and operational dependencies—for instance, attempting to use `summary_by_id` before generating fragment IDs with `fragment_context`, demonstrating incomplete understanding of the tool suite's workflow. (3) **Insufficient execution depth**: Even when correctly initiating tool usage, the model often failed to complete tasks thoroughly, with incomplete fragmentation where only partial sections were processed, leaving critical information unaddressed. These challenges underscore that effective ACM tool usage requires not just access to tools but deep understanding of efficiency trade-offs, operational dependencies, and thorough execution strategies.

**From Baseline Struggles to Systematic Guidance:** To address these inefficiencies, we crafted benchmark-specific prompts to steer tool strategies: for PI-LLM, first fragment then fold before any search or answering; for NeedleBench, coordinate `search_context` and

Table 1: Performance improvements of frontier models with ACM Tools on NeedleBench-M-RS and PI-LLM benchmarks. Both benchmarks demonstrate substantial performance gains.

| Method | NeedleBench-M-RS | | | | | PI-LLM (Update Count / Context Length) | | | | | | | |
|---|---|---|---|---|---|---|---|---|---|---|---|---|---|
| | 2-N | 3-N | 4-N | 5-N | Avg | 4/1K | 8/2K | 16/4K | 32/8K | 64/16K | 128/32K | 256/64K | Avg |
| **Claude-4-Sonnet** | | | | | | | | | | | | | |
| Baseline | 96.0 | 82.0 | 54.0 | 36.0 | 67.0 | 99.13 | 95.65 | 92.17 | 84.78 | 81.74 | 65.22 | 69.57 | 84.04 |
| w/ ACM Tools | 100.0 | 98.0 | 88.0 | 90.0 | 94.0 | 90.43 | 91.74 | 98.26 | 92.17 | 91.74 | 87.39 | 77.83 | 89.94 |
| Δ | +4.0 | +16.0 | +34.0 | +54.0 | +27.0 | -8.70 | -3.91 | +6.09 | +7.39 | +10.00 | +22.17 | +8.26 | +5.90 |
| **GPT-4.1** | | | | | | | | | | | | | |
| Baseline | 90.0 | 64.0 | 30.0 | 8.0 | 48.0 | 96.96 | 91.30 | 79.57 | 67.83 | 63.04 | 63.91 | 50.43 | 73.29 |
| w/ ACM Tools | 96.0 | 84.0 | 60.0 | 44.0 | 71.0 | 92.17 | 89.13 | 93.04 | 83.91 | 76.09 | 64.35 | 60.43 | 79.87 |
| Δ | +6.0 | +20.0 | +30.0 | +36.0 | +23.0 | -4.79 | -2.17 | +13.47 | +16.08 | +13.05 | +0.44 | +10.00 | +6.58 |
| **DeepSeek-V3** | | | | | | | | | | | | | |
| Baseline | 88.0 | 68.0 | 28.0 | 16.0 | 50.0 | 95.22 | 85.65 | 70.00 | 63.91 | 33.04 | 32.17 | 21.74 | 57.39 |
| w/ ACM Tools | 92.0 | 58.0 | 50.0 | 32.0 | 58.0 | 73.91 | 90.00 | 79.13 | 37.39 | 53.04 | 55.65 | 11.74 | 57.27 |
| Δ | +4.0 | -10.0 | +22.0 | +16.0 | +8.0 | -21.31 | +4.35 | +9.13 | -26.52 | +20.00 | +23.48 | -10.00 | -0.12 |

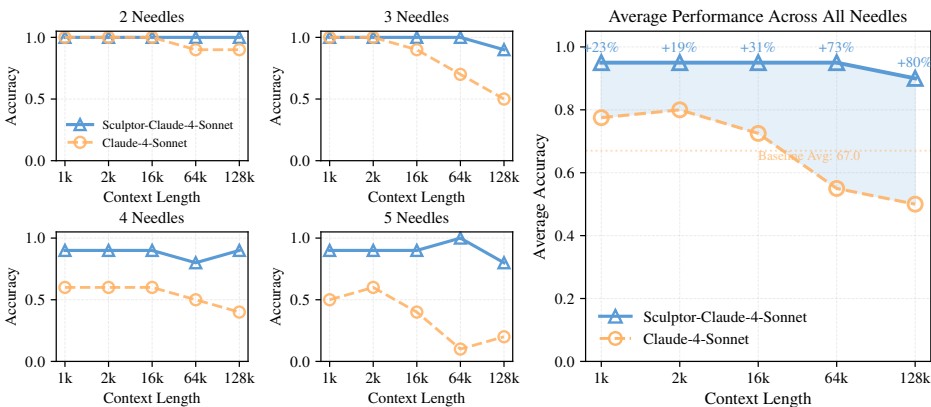

Figure 4: NeedleBench Multi-Needle Reasoning performance across different context lengths. Left: Performance by needle count showing both with tool and vanilla results. Right: Average performance across all needle counts demonstrating significant improvements.

`get_search_detail` for multi-hop retrieval. As shown in Figure 3, this guidance shifts patterns accordingly (PI-LLM: from search-heavy to fragment+fold; NeedleBench: deeper search), improving tool selection efficiency and execution completeness.

This systematic prompt engineering approach also enabled us to generate high-quality training data, collecting numerous successful tool usage examples across benchmarks. These guided patterns demonstrate that proper instruction can unlock more effective tool utilization, transforming suboptimal default behaviors into strategic, task-appropriate tool selection. The complete system prompt templates used in our experiments are provided in Appendix D.1.

**Performance Results:** Table 1 presents the evaluation results comparing models with and without ACM tools, using optimized benchmark-specific prompts. The improvements demonstrate the power of combining ACM tools with proper guidance: On NeedleBench-M-RS, Claude-4-Sonnet, GPT-4.1, and DeepSeek-V3 achieve gains of 27.0, 23.0, and 8.0 points respectively when using ACM tools with task-specific prompts, with Claude-4-Sonnet reaching 90% accuracy on 5-needle tasks. For PI-LLM, Claude-4-Sonnet and GPT-4.1 gain 5.90 and 6.58 points, while DeepSeek-V3 shows a slight decrease (-0.12), revealing persistent challenges even with prompt optimization. These results demonstrate that while prompt engineering significantly improves tool utilization, the degree of improvement varies based on each model's inherent tool-use capabilities, suggesting the need for more systematic training approaches.

Table 2: Main experimental results across benchmarks. M3 indicates our 13B baseline model without ACM tools. Sculptor-M3 is equipped with **Sculptor** tools and fine-tuned on ACM-specific data. Sculptor-M3-RL is further trained with dynamic context-aware GSPO. Similar notation applies to GLM-4.5-air models. Bold indicates best performance, underline indicates second best.

| Method | PI-LLM (Acc %) | NeedleBench-M-RS (Acc %) | MRCR (Acc %) | LongBenchV2 (Acc %) | Frames (Acc %) | Avg (Norm) |
|---|---|---|---|---|---|---|
| M3 (Baseline) | 22.5 | 30.0 | 46.3 | 33.0 | **65.2** | 39.4 |
| M3 + RAG (BM25) | 17.9 | 12.5 | 6.6 | 25.8 | 33.6 | 19.3 |
| M3 + RAG (Qwen3-Emb) | 10.9 | 13.0 | 20.6 | 29.6 | 46.0 | 24.0 |
| M3 + Mem0 | 39.2 | 19.0 | 9.2 | 29.0 | 52.8 | 29.8 |
| M3 + MemAgent | 41.5 | 24.0 | 22.1 | 29.6 | 61.5 | 35.7 |
| Sculptor-M3 | 71.8 | 67.6 | 79.1 | 29.2 | 51.2 | 59.8 |
| Sculptor-M3-RL | **99.4** | **84.8** | **85.7** | **34.5** | 64.6 | **73.8** |
| GLM-4.5-air(Baseline) | 29.4 | 24.5 | 43.1 | 46.9 | 76.0 | 44.0 |
| GLM-4.5-air + RAG (BM25) | 30.6 | 15.0 | 4.8 | 19.5 | 30.9 | 20.2 |
| GLM-4.5-air + RAG (Qwen3-Emb) | 10.9 | 12.0 | 24.1 | 27.2 | 46.6 | 24.2 |
| GLM-4.5-air + Mem0 | 18.5 | 14.5 | 6.2 | 28.8 | 62.3 | 26.1 |
| GLM-4.5-air + MemAgent | 22.2 | 17.0 | 8.6 | 33.2 | 68.6 | 29.9 |
| **Sculptor**-GLM-4.5-air | 65.2 | 58.0 | 88.5 | 31.7 | 56.7 | 60.0 |
| **Sculptor**-GLM-4.5-air-RL | **86.0** | **84.0** | **99.0** | **50.7** | **79.2** | **79.8** |

## 4.2 OPTIMIZING TOOL USE WITH REINFORCEMENT LEARNING

While prompt engineering enables effective tool usage, it requires manual effort to design task-specific prompts and still exhibits the inherent limitations discussed above. To address these challenges systematically, we employ reinforcement learning to train models that can autonomously determine optimal tool usage strategies without explicit guidance.

**Model and baselines.** We base our experiments on both an internal model and an open-source model to demonstrate the effectiveness and generalizability of our approach. Our primary model is M3, a 13B-parameter dense model that we pre-train from scratch, chosen for its strong tool-use capabilities (see Appendix C), tight compatibility with our training infrastructure, and competitive baseline performance. To validate the generalizability of our approach beyond proprietary models, we additionally evaluate GLM-4.5-air (GLM-4.5 Team et al., 2025), an open-source MoE model with 106B total parameters and 12B active parameters. In Table 2, we additionally compare three baseline approaches (all implemented on the same M3 base model for controlled comparison): retrieval-augmented generation (RAG), Mem0 (Chhikara et al., 2025) representing cross-session external memory, and MemAgent (Yu et al., 2025) as an inner working memory method. We evaluate RAG with both BM25 (keyword matching) and Qwen3-Emb (dense retrieval), representing both traditional and modern RAG approaches. Further evaluation details are provided in Appendix D.4.

**Training Data Collection.** While M3 possesses strong inherent tool-use capabilities, it requires specific training to effectively utilize the **Sculptor** tools. We generate high-quality training data through the systematic prompt engineering approach described in Section 4.1. Using Claude-4-Sonnet with carefully designed task-specific prompts, we collect successful tool usage trajectories on the BABILong (Kuratov et al., 2024) and GSM-Infinite (Zhou et al., 2025) datasets—public benchmarks featuring complex long-context reasoning challenges. This process yields diverse examples of effective ACM tool usage patterns across different task types. Combined with our conditional trajectory collection and incremental loss assignment methodology (Section 3.2), we first perform supervised fine-tuning on this data to obtain Sculptor-M3, which has learned basic ACM tool capabilities. Subsequently, we conduct RL training with dynamic context-aware GSPO on the same datasets to obtain Sculptor-M3-RL, enabling the model to autonomously discover optimal tool usage strategies. During training, we cap tool steps at 20 per turn, matching Claude-4-Sonnet's effective zero-shot usage while keeping rollouts efficient.

**Experimental Results:** Table 2 presents our experimental results. Sculptor-M3 shows improvements over baseline M3, particularly on PI-LLM (+49.3 points), NeedleBench-M-RS (+37.6 points), and MRCR (+32.8 points). After GSPO training on BABILong and GSM-Infinite datasets, Sculptor-M3-RL reaches 99.4% on PI-LLM with gains across most benchmarks. Additionally, we evaluate

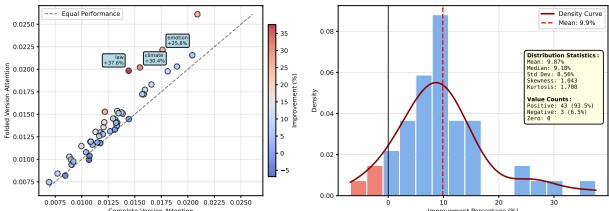 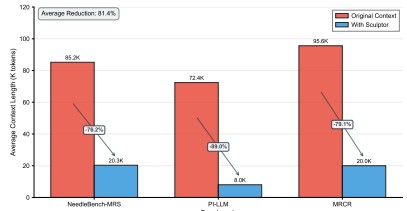

Figure 5: Value-specific attention analysis results. Left: Scatter plot comparing attention weights between folded and complete versions for 46 key-value pairs. Most points lie above the equality line, indicating improved attention with folding. Right: Distribution of attention improvements, showing a clear positive shift and confirming the systematic benefit of our approach.

Figure 6: Average context length reduction with Sculptor across benchmarks. Arrows indicate reduction percentages achieved through strategic tool usage.

GLM-4.5-air, an open-source model, demonstrating that our approach generalizes beyond our proprietary base model. GLM-4.5-air with **Sculptor** achieves substantial improvements over its baseline (+35.8 on PI-LLM, +33.5 on NeedleBench-M-RS, +45.4 on MRCR), reaching 60.0% average performance and confirming the effectiveness of our ACM tools across different model architectures. Detailed analysis of RL training dynamics and tool usage evolution is provided in Appendix G.

A critical observation from Table 2 is that **only our Sculptor approach with RL training surpasses or achieves comparable performance to the full-attention baseline** for both M3 and GLM-4.5-air. Traditional methods all fall short of their respective baselines: RAG methods introduce information loss through irreversible retrieval, achieving only 19.3% (BM25) and 24.0% (Qwen3-Emb) average for M3, and 20.2% (BM25) and 24.2% (Qwen3-Emb) for GLM-4.5-air. MemAgent's query-dependent memory accumulation discards information that appears initially irrelevant but proves critical for multi-hop reasoning—achieving 35.7% average for M3 and 29.9% for GLM-4.5-air, both below their respective baselines (39.4% and 44.0%). Mem0 underperforms MemAgent across most benchmarks, likely because it is designed for cross-session personalized memory rather than single-session long-context scenarios. The fundamental limitation is that these methods make irreversible filtering decisions based solely on the final query, without the ability to recover information that becomes relevant only after seeing subsequent context. In contrast, our ACM tools enable reversible context management—folding currently irrelevant information while preserving restoration capability. This flexibility allows Sculptor-M3-RL to reach 73.8% average (vs. 39.4% baseline) and **Sculptor**-GLM-4.5-air-RL to reach 79.8% average (vs. 44.0% baseline), demonstrating that our approach fundamentally addresses the limitations of prior methods by maintaining information accessibility rather than discarding it.

## 4.3 VALUE-SPECIFIC ATTENTION ANALYSIS

To precisely quantify how content folding impacts attention allocation to critical information, we conduct a token-level value-specific attention analysis. While traditional approaches assume that attention mechanisms naturally learn to ignore irrelevant information during pretraining, our analysis reveals that explicitly removing distracting content significantly enhances attention focus. The core idea is to measure the attention from the tokens of a specific critical value in the model's response back to the corresponding tokens of the same value in the input context. Our experiment uses 46 predefined key-value pairs from the PI-LLM benchmark as the critical information. For each pair, we calculate the attention score by first identifying the exact token positions of the value in both the input and the response. We then aggregate the attention weights across all layers and heads, averaging them to produce a single score that represents the model's focus on that specific piece of information. This allows for a direct comparison between the "folded" context and the "complete" context scenarios. The results presented in Figure 5 demonstrate a significant and systematic improvement in attention allocation. Out of 46 key-value pairs, 43 (93.5%) exhibited enhanced attention in the folded version, achieving a mean improvement of 9.87% (ranging from -6.86% to +37.56%). The scatter plot reveals a strong positive correlation ($R^2 = 0.97$) with the vast majority of data points positioned above the equality line, confirming that the improvements are consistent and not random.

Notable performance gains were observed for pairs such as "law: contract" (+37.56%), "climate: heat dome" (+30.44%), and "emotion: indifferent" (+25.83%). A one-sample t-test on the distribution of improvements confirms that they are statistically significant ($p < 0.001$), with a median improvement of 9.9%. These findings provide strong empirical evidence that folding redundant content enhances attention allocation to critical information by reducing attention dilution—even in well-pretrained models, irrelevant information interferes with attention mechanisms rather than being naturally filtered out. The consistent improvements across diverse semantic categories suggest that explicit context management through folding is more effective than relying solely on learned attention patterns.

## 4.4 Cost Analysis

To evaluate the computational efficiency of our approach, we analyze the context reduction achieved by Sculptor-M3-RL on benchmarks containing substantial irrelevant information. As shown in Figure 6, Sculptor-M3-RL achieves dramatic context reductions across these challenging benchmarks: 76.2% reduction on NeedleBench-M-RS (from 85.2K to 20.3K tokens), 89.0% on PI-LLM (from 72.4K to 8.0K tokens), and 79.1% on MRCR (from 95.6K to 20.0K tokens). These substantial reductions directly translate to computational savings, as the quadratic complexity of attention mechanisms makes processing cost heavily dependent on context length.

Importantly, our tool design minimizes additional computational overhead. Read-only tools like `search_context` preserve the prefix relationship between completions and fully reuse KV cache—they only add a few search operations while most of the context remains cached. This is similar to traditional tool use where KV cache can be efficiently reused. For context-modifying tools that do break the prefix relationship, the dramatic context reduction itself compensates for the cache invalidation cost. Processing 20K tokens even without caching is significantly faster than processing 85K tokens with full caching. This design—separating context-preserving search tools from context-modifying compression tools—ensures that our system achieves substantial context reduction with minimal computational overhead.

## 5 Limitations and Future Work

Our study primarily targets long-context scenarios, but ACM is also promising beyond long contexts. In mathematical reasoning, early mistakes can cascade due to autoregressive "prefix lock-in" that degrades subsequent correctness; folding or suppressing erroneous early steps may reset the trajectory and improve robustness (Wang & Sun, 2025; Feng et al., 2025; Wen et al., 2025). Future work will extend ACM to non-long-context domains (e.g., math, coding) and pursue richer training strategies and reward design to learn finer-grained tool-use policies, with the goal of improving performance on complex long-context benchmarks where our current results remain modest, such as LongBenchV2 and FRAMES (Bai et al., 2025; Krishna et al., 2025).

## Ethics Statement

Our research does not involve human subjects, and all experiments were conducted on publicly available benchmarks. While context manipulation tools could potentially be misused to selectively hide information, all operations in our framework are fully reversible by design, and we encourage the community to consider both the benefits and potential risks when deploying such systems.

### Acknowledgments

This work was supported by the Wuxi Research Institute of Applied Technologies, Tsinghua University under Grant 20242001120, and the Tsinghua University (AIR)-AsiaInfo Technologies (China), Inc. Joint Research Center for 6G Network and Intelligent Computing. We also thank the anonymous reviewers and the Area Chair for their insightful comments and constructive feedback that helped improve this manuscript.

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

## A    USE OF LARGE LANGUAGE MODELS

Large language models were used as a general-purpose assist tool during the writing process of this paper, primarily for grammar checking and improving clarity of technical descriptions. All scientific ideas, experimental design, and analysis were conducted by the authors. The LLMs did not contribute to research ideation or core scientific content. The authors take full responsibility for all content in this paper, including its accuracy and originality.

## B    RELATED WORK

**Long-Context Processing, Memory, and Evaluation**    Effectively processing long contexts remains a critical challenge for LLMs. Early efforts focused on expanding context windows through architectural improvements (Su et al., 2023; Chen et al., 2023; Beltagy et al., 2020) and sparse attention mechanisms (Yuan et al., 2025; Gao et al., 2025; Lu et al., 2025). Subsequently, a substantial body of work sought to further optimize performance by augmenting LLMs with external memory systems, employing comprehensive memory architectures and multi-agent frameworks to overcome context limitations (Suzgun et al., 2025; Li et al., 2025c; Yang et al., 2024; Li et al., 2025b; Wang & Chen, 2025; Chhikara et al., 2025; Packer et al., 2024; Wang et al., 2024; Yu et al., 2025). The push for longer and more complex context processing led to the development of specialized evaluation benchmarks, such as NIAH (Kamradt, 2023), NeedleBench (Li et al., 2025a), RULER (Hsieh et al., 2024a), LongBench-v2 (Bai et al., 2025), MRCR (Vodrahalli et al., 2024b), and PI-LLM (Wang & Sun, 2025). These benchmarks were instrumental in revealing that despite architectural and memory enhancements, modern LLMs still perform poorly on information-sparse tasks. Among these, work such as PI-LLM further identified a deeper reason for this phenomenon: proactive interference, where earlier information in the context disrupts the processing of later, more relevant content (Wang & Sun, 2025). These documented failures on information-sparse tasks, coupled with the diagnosis of proactive interference, provide a strong motivation for our approach of active context management. Unlike external memory solutions that focus on storage and retrieval, or sparse attention that modifies token processing patterns, our complementary method provides the model with explicit tools to selectively retain, compress, or ignore information directly within its working memory, thereby mitigating interference while operating alongside existing architectural enhancements.

**Tool-Augmented Language Models**    The integration of external tools to augment LLM capabilities is a burgeoning field of research, designed to overcome inherent model limitations such as knowledge cutoffs, hallucination, and weak mathematical reasoning. Pioneering work in this area has largely followed two paradigms. On one hand, models like Toolformer (Schick et al., 2023) demonstrate that LLMs can be fine-tuned to learn when and how to call external APIs, seamlessly incorporating their outputs into the generation process. On the other hand, prompting-based frameworks like ReAct (Yao et al., 2023b) show that LLMs can synergize chain-of-thought reasoning with tool use in a zero-shot manner, interleaving thought, action, and observation steps to solve complex tasks. Subsequent research has focused on improving the reliability and scope of tool use, with work like Gorilla (Patil et al., 2023) developing models specialized for accurate API invocation, and frameworks like ART (Paranjape et al., 2023) creating programmatic pipelines for tool-augmented multi-step reasoning. However, a common thread in this existing literature is the focus on using tools to interact with the *external* world—accessing calculators, search engines, or code interpreters. **Sculptor** diverges from this trend by proposing a novel class of tools for *internal* context management. Instead of augmenting the LLM with external knowledge, we empower it with cognitive tools to actively curate its own working memory. This positions our work as complementary to existing tool-use research. Our approach directly targets cognitive bottlenecks like proactive interference, rather than solely addressing knowledge or computational limitations. Complementary to our general active context management framework, concurrent work AgentFold (Ye et al., 2025) independently investigates a similar folding concept specifically tailored for long-horizon web agents.

**From External Compression to Internal Context Curation**    A complementary line of research focuses on reducing the computational and memory burden of long contexts through **intelligent compression** and **selection mechanisms**. The LLMLingua series (Jiang et al., 2023; 2024a; Pan et al., 2024) pioneered the use of smaller models as compressors, performing extractive compression to remove task-irrelevant sentences and phrases while preserving information density. LongLLM-

Lingua (Jiang et al., 2024a) further advanced this approach with question-aware coarse-to-fine compression and dynamic compression ratios, achieving significant improvements on long-context benchmarks. Similarly, Selective Context (Li et al., 2023) formalizes context selection as a relevance-based filtering problem for reading comprehension tasks. **At the inference level**, several methods optimize KV cache management to handle longer sequences more efficiently. StreamingLLM (Xiao et al., 2024) introduces attention sink mechanisms for online processing of extremely long inputs, while Scissorhands (Liu et al., 2023b) selectively retains only the KV pairs that will be referenced in future computations. More recent work like SnapKV (Li et al., 2024b) and KVQuant (Hooper et al., 2025) focus on pre-computation importance estimation and low-bit quantization respectively to achieve memory-efficient inference. While these compression and selection methods effectively reduce computational overhead, they share a fundamental limitation: the compression decisions are made externally to the reasoning process, either by separate models or fixed heuristics. This can lead to information loss that the primary LLM might deem crucial for its reasoning chain. In contrast, **Sculptor** enables the LLM itself to make context management decisions dynamically based on its internal reasoning state, ensuring that compression and selection align with the model's cognitive needs rather than external approximations.

**Revisable Generation and Editable Thought Processes**    Autoregressive decoding makes early errors "sticky" causing later tokens to amplify rather than fix them. Proactive interference tests show that retrieval accuracy degrades as semantically related but obsolete updates accumulate, underscoring the cost of an immutable context (Wang & Sun, 2025). Causally, pruning failed reasoning branches—or removing their surface forms from the visible history—immediately improves subsequent correctness, indicating the harmful persistence of erroneous traces (Feng et al., 2025). To mitigate this prefix lock-in, one line of work explores parallel or branched reasoning, such as the search-based Tree of Thoughts and the native parallelism in ParaThinker, which reduces "tunnel vision" at a small latency overhead (Yao et al., 2023a; Wen et al., 2025). A more fundamental approach alters the generation process itself, making outputs inherently revisable. This includes models that perform discrete edits, like iterative refinement via masking (Ghazvininejad et al., 2019) or sequence modification through insertion and deletion operations (Stern et al., 2019; Gu et al., 2019). Another family of non-autoregressive paradigms, such as diffusion LMs, enables global backtracking by denoising entire sequences in parallel (Li et al., 2022; Austin et al., 2021). Complementing these architectural shifts, test-time self-revision loops like Self-Refine and Reflexion demonstrate that lightweight edits to intermediate outputs reliably improve final solutions (Madaan et al., 2023; Shinn et al., 2023). Collectively, these findings build a strong case for equipping models with mechanisms to remove, rewrite, or compress their working context during reasoning—rather than merely appending tokens—so they can correct course instead of being trapped by early errors.

## C  BASELINE MODEL PERFORMANCE

We evaluate M3 across standard benchmarks and compare it with both smaller-scale models (Qwen3-8B, Qwen3-14B) and frontier models to establish baseline capabilities before ACM training. The results are presented in Table 3. Despite being a 13B model, M3 demonstrates exceptional tool-use performance, achieving 61.0% on Tau2-retail (Barres et al., 2025) (vs. 27.9% for Qwen3-8B), 61.8% on AceBench (Chen et al., 2025) (vs. 24.3% for Qwen3-8B), and 32.0% on SWE-bench Verified (vs. 3.3% for Qwen3-8B). This strong tool-calling foundation makes it particularly suitable for demonstrating ACM effectiveness. Additionally, we include GLM-4.5-Air benchmark scores in Table 3 to provide comprehensive baseline comparisons for the open-source model used in our main experiments (Table 2). Benchmark results for GLM-4.5-Air are reported from their technical report (GLM-4.5 Team et al., 2025). Benchmark results for DeepSeek-V3, GPT-4.1, and Claude-4-Sonnet (excluding NeedleBench-MRS and PI-LLM) are taken from the Kimi-K2 technical report (Team et al., 2025).

For Qwen3-8B and Qwen3-14B models, we followed the official documentation[1] to enable 128K context length support through RoPE scaling (Su et al., 2023) with the YaRN method (Peng et al., 2023), using a scaling factor of 4.0 to extend from their original 32K context window to 128K tokens. This configuration was necessary for fair comparison on long-context benchmarks.

---

[1]https://qwen.readthedocs.io/en/latest/deployment/vllm.html

Table 3: Performance of M3 (13B parameters) compared to other models on standard benchmarks. Left: smaller-scale models (8B-14B). Right: frontier models. M3 demonstrates particularly strong tool-use capabilities (Tau2, AceBench) and coding performance (SWE-bench Verified).

| Benchmark | Qwen3-8B | Qwen3-14B | M3 | GLM-4.5-Air | Kimi-K2 | DeepSeek-V3 | GPT-4.1 | Claude-4-Sonnet |
|---|---|---|---|---|---|---|---|---|
| **Coding Tasks** | | | | | | | | |
| LiveCodeBench v6 (Pass@1) | 50.2 | 51.8 | 25.1 | 70.7 | 53.7 | 46.9 | 44.7 | 48.5 |
| MultiPL-E (Pass@1) | 70.4 | 77.0 | 72.4 | – | 85.7 | 83.1 | 86.7 | 88.6 |
| SWE-bench Verified (Pass@1) | 3.3 | 5.8 | 32.0 | 57.6 | 51.8 | 36.6 | 40.8 | 50.2 |
| **Tool Use Tasks** | | | | | | | | |
| Tau2 retail (Avg@4) | 27.9 | 36.2 | 61.0 | 77.9 | 70.6 | 69.1 | 74.8 | 75.0 |
| Tau2 airline (Avg@4) | 18.0 | 39.0 | 54.0 | 60.8 | 56.5 | 39.0 | 54.5 | 55.5 |
| AceBench (Acc.) | 24.3 | 23.7 | 61.8 | 76.4 | 76.5 | 72.7 | 80.1 | 76.2 |
| **Math & STEM Tasks** | | | | | | | | |
| MATH-500 (Acc.) | 92.2 | 95.4 | 80.2 | 98.1 | 97.4 | 94.0 | 92.4 | 94.0 |
| AIME 2024 (Avg@64) | 60.9 | 60.8 | 17.7 | 89.4 | 69.6 | 59.4 | 46.5 | 43.4 |
| GPQA-Diamond (Avg@8) | 53.0 | 58.1 | 45.2 | 75.0 | 75.1 | 68.4 | 66.3 | 70.0 |
| **General Tasks** | | | | | | | | |
| MMLU (EM) | 80.1 | 85.0 | 78.6 | 87.4 | 89.5 | 89.4 | 90.4 | 91.5 |
| MMLU-Pro (EM) | 74.5 | 77.5 | 65.2 | 81.4 | 81.1 | 81.2 | 81.8 | 83.7 |
| IFEval (Prompt Strict) | 34.9 | 35.5 | 77.1 | 86.3 | 89.8 | 81.1 | 88.0 | 87.6 |
| SimpleQA (Correct) | 6.7 | 8.8 | 7.4 | 14.5 | 31.0 | 27.7 | 42.3 | 15.9 |

# D    EVALUATION DETAILS

## D.1    BENCHMARK-SPECIFIC SYSTEM PROMPTS

As described in Section 4.1, we employed prompt engineering to enhance tool utilization capabilities across frontier models. The system prompts presented here are the final optimized versions used in our zero-shot evaluation for PI-LLM and NeedleBench benchmarks. These benchmark-specific prompts significantly improved Claude-4-Sonnet's performance, demonstrating how targeted prompt optimization can unlock more effective **Sculptor** tool usage patterns.

---

**System Prompt for PI-LLM Benchmark**

**System Prompt:**
You are an intelligent assistant specialized for PI-LLM (Proactive Interference) testing. Your task is to track continuous updates to multiple key-value pairs and accurately remember the latest value for each key amidst substantial interference information.
Remember: First use the fragment_context tool to split the long text into multiple fragments, then use fold_fragment to fold unimportant, earlier key-value updates, allowing you to concentrate on the final updates. The recommended approach is to divide the entire update stream into multiple fragments (e.g., ten fragments), then keep only the last two or three fragments while folding the rest. This strategy enables focus on the current, most recent content without being distracted by earlier information.

---

Figure 7: System prompt for PI-LLM benchmark, designed to handle proactive interference through strategic tool usage.

## D.2    UNIFIED SYSTEM PROMPT

The unified system prompt is used in our initial zero-shot evaluations and RL training experiments. This minimal, general-purpose prompt provides only basic guidance about available capabilities without prescriptive task-specific strategies. As described in Section 4.1, experiments with this unified prompt revealed inherent challenges of unguided tool use, including suboptimal tool selection patterns and insufficient execution depth. During RL training, the same prompt enables the model to autonomously discover optimal tool usage patterns across diverse contexts, as shown in Figure 9.

This approach ensures that the model learns generalizable context management strategies rather than memorizing task-specific patterns, leading to more robust performance across diverse long-context.

---

**System Prompt for NeedleBench Multi-Needle Reasoning**

**System Prompt:**
You are an agent skilled at analyzing family relationships between different people. You have "search_context" and "get_search_detail" tools. You excel at conducting chained searches for key information in long texts until you find complete information to reach your desired final answer.

When searching for the oldest ancestor, ensure that every person name found has been verified through the search tools to confirm they truly have no higher-level ancestors before concluding your reasoning.

---

Figure 8: System prompt for NeedleBench Multi-Needle Reasoning, optimized for multi-hop retrieval tasks.

---

**Unified System Prompt**

**System Prompt:**
You are a helpful assistant. You can autonomously manage your own context: fold irrelevant information, focus on useful details, summarize long texts to keep your context concise, and use search tools to find key information in large documents.

---

Figure 9: The unified general-purpose prompt used both in initial zero-shot evaluations (to understand natural tool interaction patterns) and in RL training (to enable autonomous learning of tool usage strategies without prescriptive guidance).

### D.3   BENCHMARK DETAILS

We provide detailed configurations for the benchmarks used in our experiments:

**NeedleBench Multi-Needle Reasoning:**   For efficiency while maintaining representativeness, we test with a fixed depth of 40%, as our tool-based approach shows minimal sensitivity to needle position within the context. We examine context lengths of 1k, 2k, 16k, 64k, and 128k tokens, with each configuration evaluated across 10 runs per dataset to ensure statistical significance. The multi-needle variant requires connecting 2, 3, 4, and 5 needles simultaneously, making it substantially more challenging than single-needle retrieval tasks.

**Data Processing for Context Length Constraints:**   To ensure evaluation within our model's 128k context window, we apply minimal preprocessing. For MRCR, we filter out test samples exceeding 128k tokens. For LongBench v2, we truncate samples exceeding 128k tokens using our tokenizer.

### D.4   BASELINE METHOD EVALUATION DETAILS: RAG, MEM0, AND MEMAGENT

We include three baseline approaches for long-context processing in our main results (see Table 2): retrieval-augmented generation (**RAG**) as a traditional long-context method, **Mem0** as a cross-session external memory system, and **MemAgent** as an inner working memory approach. All baselines are evaluated under a unified, lightweight interface that accepts plain strings or standard message arrays, without dataset-specific restructuring.

For **RAG**, we evaluate two retrieval approaches to represent both traditional and modern methods:

**BM25-based RAG**[2]: We adopt a BM25-only pipeline aligned with LongBench-style retrieval. The input is sentence-split with the same punctuation and length heuristics as common LongBench implementations, then chunked at 200 tokens. A pseudo query is formed by concatenating the first and last 500 tokens of the full context when an explicit query is not provided. Chunks are ranked

---

[2]https://github.com/THUDM/LongBench/tree/main/LongBench/retrieval/BM25

by BM25 and concatenated from high to low until the accumulated length reaches $\approx 1500$ tokens. The system prompt constrains the model to answer strictly based on the retrieved context. This BM25-only design avoids external dense embeddings and evaluates the model's intrinsic ability to reason over the retrieved snippets.

**Embedding-based RAG**: We additionally evaluate dense retrieval using the Qwen3 embedding model (Zhang et al., 2025b), a state-of-the-art text embedding model. The input is chunked similarly at 200 tokens. For each chunk, we compute dense embeddings using the Qwen3 embedding model and rank chunks by cosine similarity to the query embedding. The top-ranked chunks are concatenated until reaching $\approx 1500$ tokens, and the model generates answers based on the retrieved context. This approach represents modern embedding-based RAG systems widely adopted in production settings.

For **MemAgent**(Yu et al., 2025)[3], we follow their implementation with iterative memory updates. Extremely long inputs are first symmetrically trimmed to a maximum visible length of about $120\,\mathrm{k}$ tokens to avoid one-sided truncation. The remaining text is processed in fixed $5\,\mathrm{k}$-token chunks. At each step the model updates an explicit "memory" that preserves previously useful information and integrates newly relevant details from the current chunk; the final answer is generated using the last memory along with the query. When no explicit query is given, we construct a short pseudo query from the first and last $500$ tokens of the source. Unless otherwise noted, defaults are: max context length $\approx 120\,\mathrm{k}$ tokens, chunk size $5\,\mathrm{k}$ tokens, and maximum generation length $1024$ tokens.

For **Mem0**(Chhikara et al., 2025), we adapt the official implementation for long-context evaluation. Since Mem0 is designed for cross-session user preference memory rather than single-session document processing, we modify: (1) the fact extraction prompt to focus on document-relevant information instead of user preferences, and (2) the chunking mechanism to include the query in each $5\,\mathrm{k}$-token chunk, ensuring the memory LLM can extract question-relevant facts. Each question uses a unique user ID to prevent memory interference across samples.

These choices emphasize reproducibility and represent diverse approaches to long-context processing: traditional and modern retrieval methods for RAG, cross-session external memory for Mem0, and inner working memory for MemAgent. Detailed hyperparameters are reflected in the text above rather than bespoke configuration tables to keep the protocol concise and focused.

## E  DYNAMIC CONTEXT-AWARE TRAINING DATA COLLECTION

Algorithm 1 presents our conditional trajectory collection algorithm with incremental loss assignment for dynamic context-aware RL training. The algorithm identifies context-modifying tool calls and creates separate training instances at each modification point, with incremental loss assignment to prevent redundant learning across multiple trajectory snapshots.

## F  TRAINING CONFIGURATION

To ensure reproducibility and facilitate future research building upon our work, we provide the detailed training hyperparameters and hardware configuration used for GSPO training in Table 4. These settings represent the optimal configuration determined through extensive experimentation for training Sculptor-M3-RL with dynamic context-aware capabilities.

Table 4: GSPO training configuration.

| Training Hyperparameters | | Hardware & Parallelism | |
| --- | --- | --- | --- |
| Learning rate | $1 \times 10^{-6}$ | GPU type | NVIDIA H800 (80GB) |
| Training iterations | 200 | Total GPUs | 128 (64 train, 64 rollout) |
| Clip ratio (lower) | 0.0003 | Tensor parallel (TP) | 1 |
| Clip ratio (upper) | 0.0004 | Pipeline parallel (PP) | 4 |
| KL penalty ($\alpha$) | 0.0 | Context parallel (CP) | 16 |
| LM regularization | 0.1 | Data parallel (DP) | 4 |
| Optimizer | AdamW | Max sequence length | 128k tokens |

---

[3]`https://github.com/BytedTsinghua-SIA/MemAgent/blob/main/quickstart.py`

---

**Algorithm 1** Conditional Trajectory Collection with Incremental Loss Assignment

---

**Require:** Initial query $Q$, complete interaction trajectory with assistant completions $\{C_i\}_{i=0}^{n}$ and tool results $\{T_i\}_{i=0}^{n-1}$.
**Require:** Set of context-modifying tools $\mathcal{T}_{ctx}$ (fragment, fold, summarize, restore operations).
**Ensure:** Training dataset $\mathcal{D}_{train}$ containing (trajectory, loss_mask) pairs.
1: Initialize $\mathcal{D}_{train} \leftarrow \emptyset$
2: Initialize $trained\_indices \leftarrow \emptyset$    ▷ Track completion indices that have been assigned loss=1
3: **for** $i = 0$ **to** $n$ **do**
4:   Extract tool call $a_i$ from completion $C_i$
5:   **if** $a_i \in \mathcal{T}_{ctx}$ **or** $i = n$ **then**       ▷ Context modification or final completion
6:                 ▷ Create trajectory snapshot up to current point
7:    $trajectory \leftarrow [Q, C_0, T_0, C_1, T_1, \ldots, C_i]$
8:    **if** $i < n$ **then**
9:     $trajectory \leftarrow trajectory + [T_i]$      ▷ Include tool result if not final
10:    **end if**
11:                ▷ Create incremental loss mask
12:    Initialize $loss\_mask$ with zeros for all elements in $trajectory$
13:    **for** $j = 0$ **to** $i$ **do**
14:     **if** $j \notin trained\_indices$ **then**    ▷ Only assign loss to new completions
15:      $loss\_mask[C_j] \leftarrow 1$       ▷ Enable loss for completion $C_j$
16:      $trained\_indices \leftarrow trained\_indices \cup \{j\}$
17:     **end if**
18:    **end for**
19:    $\mathcal{D}_{train} \leftarrow \mathcal{D}_{train} \cup \{(trajectory, loss\_mask)\}$
20:    **if** $a_i \in \mathcal{T}_{ctx}$ **then**      ▷ Update query for next iteration if context modified
21:     $Q \leftarrow$ ApplyToolEffect$(Q, a_i, T_i)$     ▷ Apply context modification
22:    **end if**
23:   **end if**
24: **end for**
25: **return** $\mathcal{D}_{train}$

---

**Reward Design.**    Our reward function for GSPO training is defined as:

$$r(x, \tau) = \begin{cases} 1 & \text{if correct answer} \\ -1 & \text{if format error or } n_{\text{tools}} > 20 \text{ or } |\tau| > 128\text{k tokens} \\ 0 & \text{otherwise} \end{cases} \quad (4)$$

This design encourages correct task completion while penalizing excessive tool usage, overly long trajectories, and malformed outputs.

# G    RL TRAINING DYNAMICS AND TOOL USAGE ANALYSIS

This section provides detailed analysis of the reinforcement learning training process, including training dynamics across iterations and the evolution of tool usage patterns.

## G.1    TRAINING ACCURACY COMPARISON: GSPO VS GRPO

We compare two RL training approaches: GSPO (Group Sequence Policy Optimization) (Zheng et al., 2025) adapted with our dynamic context-aware extensions, and GRPO (Group Relative Policy Optimization) (Shao et al., 2024) as a baseline. Figure 10 shows the training accuracy curves across RL iterations. Both methods demonstrate consistent learning progress, achieving comparable final performance ( 66-67%). GSPO shows slightly more stable convergence during later training iterations, validating our choice for the main experiments. The similar performance of both algorithms suggests that our dynamic context-aware training framework is not tightly coupled to a specific RL algorithm choice, demonstrating compatibility with different policy optimization methods. This indicates that when given the same training data, the choice of specific RL algorithm has limited impact on final results, with data quality and task design being more critical factors.

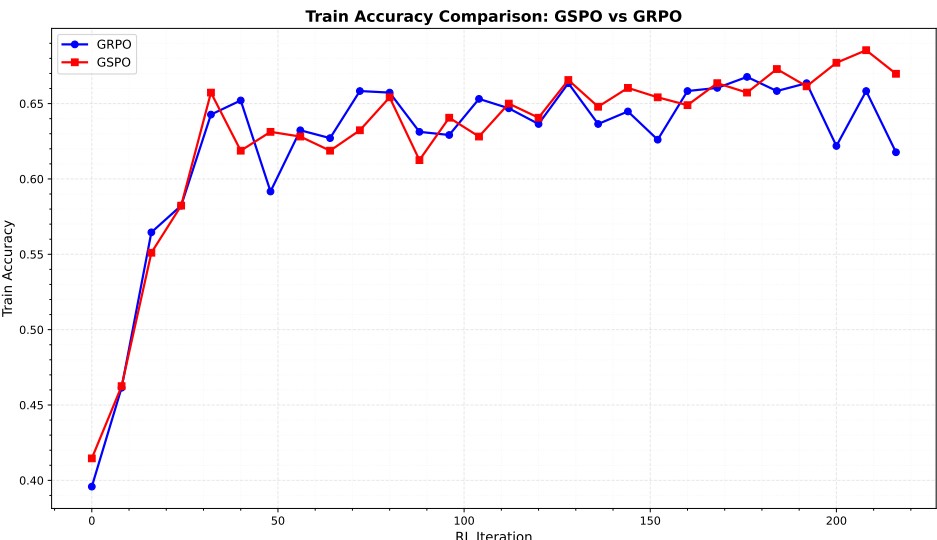

Figure 10: Training accuracy comparison between GSPO and GRPO across RL iterations. Both methods show steady improvement and achieve similar final performance.

## G.2    TOOL USAGE EVOLUTION DURING RL TRAINING

To understand how the RL policy learns to use different tools, we analyze tool usage patterns across training iterations. We examine both the overall diversity of tool usage (measured by entropy) and specific tool call patterns on PI-LLM tasks, which require extensive context management through folding and selective restoration of information.

### G.2.1 TOOL USAGE DIVERSITY AND INTENSITY

As RL training progresses, we observe two complementary trends in tool usage behavior. First, tool usage entropy increases substantially (training set: 1.17 to 2.09, +77.6%; evaluation set: 1.69 to 1.98, +17.0%), indicating the policy learns to leverage a more diverse repertoire of tools based on task requirements. Second, the average number of tool calls per task increases from 7.26 to 8.47 (+16.6%) on evaluation benchmarks, demonstrating more intensive and sophisticated use of available tools. Together, these patterns show that the policy evolves from relying on a few dominant tools to strategically employing the full range of capabilities (fold, restore, search, etc.) when needed, which is critical for robust performance across diverse long-context scenarios.

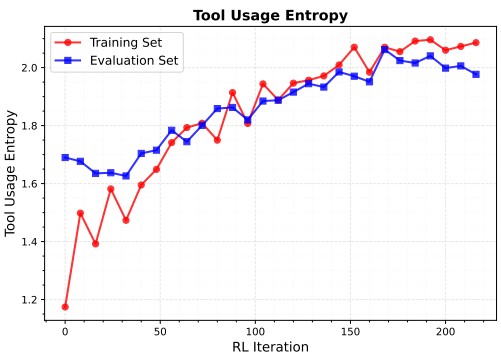

Figure 11: Tool usage entropy across RL training iterations. The substantial increase indicates more diverse tool selection.

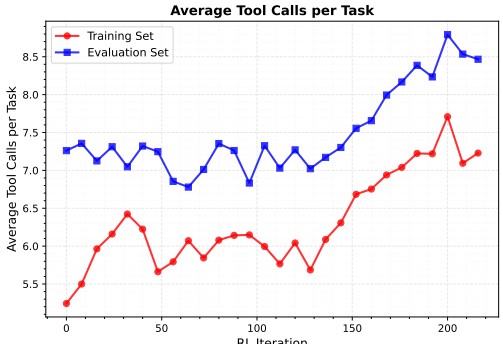

Figure 12: Average tool calls per task across RL training iterations.

### G.2.2 TOOL CALL PATTERNS ON PI-LLM TASKS

Figure 13 presents how the RL policy learns to balance compression and information preservation on PI-LLM tasks. Initially, the policy relies heavily on `fold_fragment` for context compression with minimal use of `restore_fragment` (0.052 calls per task). However, excessive folding can discard critical information, leading to accuracy degradation. Through RL training with accuracy-based rewards, the policy gradually learns to recover over-compressed useful information. The usage of `fold_fragment` increases from 4.85 to 7.08 calls per task (+45.8%), while critically, `restore_fragment` usage grows from 0.052 to 0.267 calls per task (+416.6%). More significantly, comparing early iterations (0-40) to late iterations (160-208), the average restore usage increases from 0.0746 to 0.2068, representing a **177.2% growth**. Notably, `summarize_fragment` remains near zero throughout training, indicating the policy intelligently recognizes that PI-LLM tasks involve truly irrelevant context that should be directly folded rather than summarized—a more efficient strategy when the discarded information is genuinely unnecessary. This learned restore behavior enables the model to achieve near-perfect performance on PI-LLM tasks, demonstrating that the RL policy successfully learns to proactively recover information when aggressive compression risks losing critical details.

The substantial increase in restore usage during later training iterations directly addresses concerns about robustness to over-compression. Early in training, the policy predominantly uses folding without restoration, which can lead to information loss and accuracy drops. The accuracy-based reward signal drives the policy to discover that restoring over-compressed fragments can recover lost information and improve task performance. The simultaneous increase in both fold and restore usage indicates the policy learns to compress aggressively while developing safeguards through selective restoration. These patterns validate that our reversible tool design successfully enables the RL policy to learn robust context management strategies, ultimately achieving near-perfect accuracy on PI-LLM by balancing compression efficiency with the ability to recover critical information when needed.

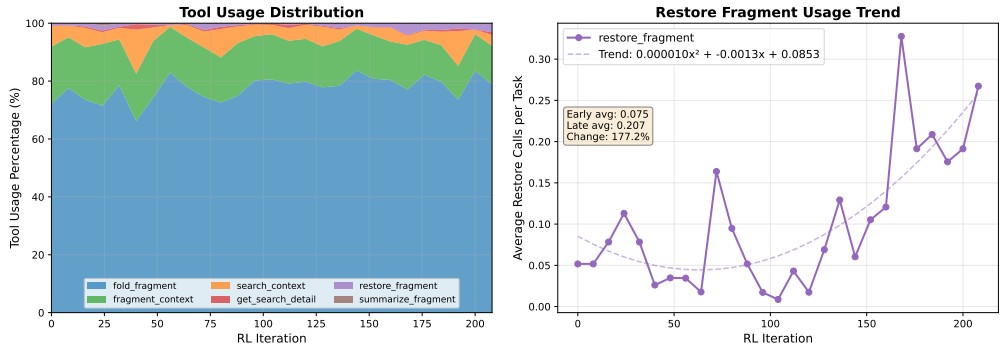

Figure 13: Tool usage evolution on PI-LLM tasks across RL training iterations. (Left) Percentage distribution showing restore_fragment's relative proportion increasing over training, demonstrating more diverse tool usage. (Right) Absolute restore usage trend showing the average restore calls per task steadily increases, indicating the policy learns to proactively recover information when needed.

## H  SCULPTOR TOOL SUITE SCHEMAS

We provide the complete JSON schemas for all six core **Sculptor** tools, detailing their parameters and usage specifications. When tools like `fold_fragment` or `summarize_fragment` modify context content, the original text is temporarily stored in memory to enable complete restoration via `restore_fragment`. This ensures no information is permanently lost during context management operations.

```json
{
  "type": "function",
  "function": {
    "name": "fragment_context",
    "description": "Fragment conversation content between specified
        markers into manageable pieces. Useful for breaking down long
        text sections for detailed analysis.",
    "parameters": {
      "type": "object",
      "properties": {
        "start_marker": {
          "type": "string",
          "description": "Start marker text to identify the beginning of
              content to fragment"
        },
        "end_marker": {
          "type": "string",
          "description": "End marker text to identify the end of content
              to fragment"
        },
        "num_fragments": {
          "type": "integer",
          "default": 5,
          "minimum": 1,
          "maximum": 20,
          "description": "Number of fragments to create (default: 5)"
        },
        "role": {
          "type": "string",
          "enum": ["user", "assistant", "all"],
          "default": "user",
          "description": "Which role's messages to search in (default:
              user)"
        }
      },
      "required": ["start_marker", "end_marker"],
      "additionalProperties": false
    }
  }
}
```

Figure 14: JSON schema for `fragment_context` tool: Fragments conversation content between markers.

```
{
  "type": "function",
  "function": {
    "name": "fold_fragment",
    "description": "Fold (hide) a conversation fragment to reduce visible
        context length. The content is preserved and can be expanded
      later.",
    "parameters": {
      "type": "object",
      "properties": {
        "fragment_id": {
          "type": "string",
          "description": "ID of the fragment to fold (e.g., 'f1a2b3')"
        }
      },
      "required": ["fragment_id"],
      "additionalProperties": false
    }
  }
}
```

Figure 15: JSON schema for `fold_fragment` tool: Hides fragments to reduce context.

```
{
  "type": "function",
  "function": {
    "name": "restore_fragment",
    "description": "Restore a fragment to its original content from ACM
        storage. Works for both summarized and folded fragments.",
    "parameters": {
      "type": "object",
      "properties": {
        "fragment_id": {
          "type": "string",
          "description": "ID of the fragment to restore (e.g., 'f1a2b3')"
        }
      },
      "required": ["fragment_id"],
      "additionalProperties": false
    }
  }
}
```

Figure 16: JSON schema for `restore_fragment` tool: Restores modified fragments.

```
{
  "type": "function",
  "function": {
    "name": "summarize_fragment",
    "description": "Summarize a conversation fragment using LLM to
        compress content while preserving key information. Supports focus
        -oriented summarization.",
    "parameters": {
      "type": "object",
      "properties": {
        "fragment_id": {
          "type": "string",
          "description": "ID of the fragment to summarize (e.g., 'f1a2b3
              ')"
        },
        "focus": {
          "type": "string",
          "description": "Focus area for the summary (e.g., 'technical
              details', 'key decisions', 'action items', 'main points', '
              problems', 'solutions')"
        }
      },
      "required": ["fragment_id", "focus"],
      "additionalProperties": false
    }
  }
}
```

Figure 17: JSON schema for `summarize_fragment` tool: Compresses fragments with LLM.

```json
{
  "type": "function",
  "function": {
    "name": "search_context",
    "description": "Search tool for finding exact text matches in
        conversation history.",
    "parameters": {
      "type": "object",
      "properties": {
        "query": {
          "type": "string",
          "description": "Exact text to search for in conversation
              history"
        },
        "role": {
          "type": "string",
          "enum": ["user", "assistant", "all"],
          "default": "user",
          "description": "Filter by message role (default: user)"
        },
        "max_results": {
          "type": "integer",
          "default": 10,
          "minimum": 1,
          "maximum": 50,
          "description": "Maximum number of results to return"
        },
        "context_size": {
          "type": "integer",
          "default": 200,
          "minimum": 50,
          "maximum": 1000,
          "description": "Context characters before/after match"
        }
      },
      "required": ["query"],
      "additionalProperties": false
    }
  }
}
```

Figure 18: JSON schema for `search_context` tool: Exact text search in conversation.

```
{
  "type": "function",
  "function": {
    "name": "get_search_detail",
    "description": "Get detailed context for a search result by its ID.
        Retrieves extended context around the search match position.",
    "parameters": {
      "type": "object",
      "properties": {
        "search_id": {
          "type": "string",
          "description": "Search result ID from search_context (e.g., '
              s1a2b3')"
        },
        "extended_context": {
          "type": "integer",
          "default": 500,
          "minimum": 100,
          "maximum": 2000,
          "description": "Number of characters to show before and after
              the match (default: 500)"
        }
      },
      "required": ["search_id"],
      "additionalProperties": false
    }
  }
}
```

Figure 19: JSON schema for `get_search_detail` tool: Retrieves extended context.

