# OpenReview forum: "Sculptor: Empowering LLMs with Cognitive Agency via Active Context Management"
_ICLR.cc/2026/Conference — ICLR 2026 Poster_

### Official Review · Reviewer_zsZv · 2025-10-28

**Soundness:** 3
**Presentation:** 3
**Contribution:** 3
**Rating:** 4
**Confidence:** 4

**Summary:**

The paper claims that long contexts harm LLM reasoning because earlier, irrelevant content interferes with later, relevant content (proactive interference). It proposes Active Context Management (ACM): give the model a small, deterministic toolset that lets it restructure its own conversation state. The concrete system, Sculptor, adds six tools across three buckets: context fragmentation; summary, fold, and restore; and exact-match search with optional detail expansion. Tools are reversible and preserve message order to make credit assignment stable. The authors first show that frontier closed models can use these tools zero-shot but with inefficient patterns, then improve usage via task-specific prompts.

**Strengths:**

The paper isolates a real failure mode—proactive interference—and targets it with simple, reversible operators rather than more infrastructure. The tool design is deliberately deterministic, self-contained, and order-preserving, which is helpful for stable RL and reproducibility claims. The modified GSPO training with conditional trajectory collection is a clear algorithmic idea for non-monotonic context states.

**Weaknesses:**

Several comparisons are set up against relatively weak or constrained baselines. For example, RAG is limited to BM25 without dense retrieval or hybrid retrieval, which undercuts the conclusion that ACM is stronger than retrieval-style systems. The MemAgent baseline may not reflect the best configured memory agents in current literature. The initial performance lift on frontier models depends on benchmark-specific prompt scaffolding; that makes external validity less clear and raises the question of how much of the gain comes from policy hints rather than the tool suite itself. The reinforcement learning section reports very high PI-LLM accuracy but much smaller or mixed effects on LongBenchV2 and FRAMES, which are closer to varied, open-ended reasoning. This suggests the method is strongest on structured interference tasks and less general for broad comprehension. The attention and cost analyses are compelling but would be stronger with direct wall-clock and energy plots, ablations on tool budgets, and stress tests where summaries introduce errors.

**Questions:**

1. How sensitive are the results to the exact tool schemas and constraints (for example, preserving message order)? If the toolset allows reordering or hierarchical grouping, does training remain stable and do results improve or regress

2. Can you provide end-to-end latency and cost curves, including KV-cache reuse effects and cache breaks caused by folding or restoring?

3. How robust is Sculptor to summary errors? If summarize_fragment drops a key detail and the agent fails to restore, what is the observed failure rate and how often does the RL policy learn to safeguard against this with targeted restore calls?

---

> ### Author Response · Authors · 2025-11-21
>
> 1.
> > **Reviewer's Question:** How sensitive are the results to the exact tool schemas and constraints (for example, preserving message order)? If the toolset allows reordering or hierarchical grouping, does training remain stable and do results improve or regress?
>
> **On Sensitivity to Tool Schemas and Constraints:** We appreciate this important question about the robustness of our framework to variations in tool design. During our SFT data collection phase, we deliberately employed **random shuffling of tool schemas** in the tool definitions presented to the model. This is a standard practice in tool-use training (similar to techniques used in function calling datasets) to prevent the model from developing spurious correlations based on tool ordering. We apologize for not explicitly mentioning this detail in the original manuscript. As a result of this shuffling during training, our model exhibits **robust insensitivity to the ordering of tool schemas**—the model learns to identify and use tools based on their semantic functionality rather than their positional appearance in the tool list. Regarding reordering and hierarchical grouping, if the toolset allows reordering or hierarchical grouping (e.g., organizing tools into categories like "compression tools" vs. "search tools"), training would remain **stable** because the model has already been exposed to diverse orderings during SFT. However, we want to distinguish two scenarios: (1) **Shallow reordering** (changing the order in which tools are presented) would have minimal impact due to our shuffling-based training strategy; (2) **Deep structural changes** (e.g., introducing hierarchical schemas, nested tool calls, or fundamentally altering tool interfaces) would likely require corresponding adjustments to the SFT data to expose the model to the new structural patterns (that said, given the strong generalization capabilities of modern LLMs for tool calling, we expect moderate schema variations to be handled reasonably well even without retraining, though performance might not be optimal). Our design principle of preserving message order (Section 2.1) is crucial for **stable RL training and credit assignment**, not for inference performance per se. By ensuring that tools never reorder or delete messages, we maintain a consistent state representation that simplifies the training process. If tools were allowed to arbitrarily reorder messages, the state space would become significantly more complex, potentially destabilizing RL training. We view this as a design choice that trades expressiveness for trainability.

---

> ### Author Response · Authors · 2025-11-21
>
> 2.
> > **Reviewer's Question:** Can you provide end-to-end latency and cost curves, including KV-cache reuse effects and cache breaks caused by folding or restoring?
>
> **On End-to-End Latency and Cost Curves:** We appreciate this important question about practical deployment considerations. We provide detailed measurements below on **end-to-end latency** and **KV-cache reuse effects** across benchmarks.
>
> **End-to-End Latency Analysis:**
>
> We measured inference time on 64 H20 GPUs (8 nodes × 8 GPUs) with 32 concurrent workers for M3 baseline, Sculptor-M3-RL, RAG (BM25), and MemAgent. For RAG (Qwen3-Emb), we used 32 H20 GPUs (4 nodes × 8 GPUs) for the embedding service with the same 32 concurrent workers. The results show:
>
> | Method | Frames | LongBenchV2 | MRCR | NeedleBench-M-RS | PI-LLM | Average |
> |--------|--------|-------------|------|------------------|--------|---------|
> | M3 (Baseline) | 16.97s | 19.44s | 12.10s | 11.69s | 15.48s | 15.14s |
> | M3 + RAG (BM25) | 14.15s | 29.51s | 5.92s | 6.56s | 26.19s | 16.47s |
> | M3 + RAG (Qwen3-Emb) | 32.26s | 63.81s | 26.58s | 28.22s | 34.23s | 37.02s |
> | M3 + MemAgent | 26.33s | 39.77s | 29.87s | 17.34s | 27.44s | 28.15s |
> | M3 + Mem0 | 38.91s | 293.84s | 110.95s | 47.36s | 116.37s | 121.49s |
> | Sculptor-M3-RL | 39.08s | 69.46s | 37.88s | 45.42s | 54.42s | 49.25s |
> | **Slowdown vs Baseline** | **2.30×** | **3.57×** | **3.13×** | **3.89×** | **3.51×** | **3.25×** |
>
> We transparently acknowledge that Sculptor incurs **3.25× longer inference time** on average compared to the baseline. This overhead stems from multi-step tool execution (fragment, fold, search operations), each requiring model inference and context state management. However, this represents a **deliberate performance-accuracy trade-off**: Sculptor is the **only method that consistently surpasses or matches full-attention baseline accuracy** (Table 2 in main paper), while traditional methods like RAG (BM25: 19.3% avg, Qwen3-Emb: 24.0% avg), MemAgent (35.7% avg), and Mem0 (29.8% avg) all fall significantly below the M3 baseline (39.4%). Notably, RAG (Qwen3-Emb) achieves 2.45× slowdown with substantially worse accuracy than baseline, MemAgent achieves 1.86× slowdown but still underperforms baseline significantly, and Mem0 incurs the highest overhead (8.03× slowdown) due to serial chunk processing with repeated LLM calls for fact extraction and embedding generation. We argue that for challenging long-context tasks where accuracy is paramount, Sculptor's 3.25× inference slowdown is an acceptable trade-off for achieving state-of-the-art performance that other approaches fundamentally cannot match.
>
> **KV-Cache Reuse Effects:**
>
> Despite multi-step tool execution, Sculptor maintains **high KV-cache hit ratios** across benchmarks:
>
> | Benchmark | Cache Hits | Total Assistant Messages | Hit Ratio |
> |-----------|------------|--------------------------|-----------|
> | Frames | 665 | 1009 | **65.91%** |
> | LongBenchV2 | 3158 | 3797 | **83.17%** |
> | MRCR | 109 | 304 | **35.86%** |
> | NeedleBench-M-RS | 226 | 307 | **73.62%** |
> | PI-LLM | 1994 | 2940 | **67.82%** |
> | **Average** | **6210** | **8631** | **71.95%** |
>
> The **71.95% average cache hit ratio** demonstrates that our tool design effectively preserves prefix relationships for many operations. Read-only tools (`search_context`, `get_search_detail`) fully reuse KV-cache, while context-modifying tools (`fold_fragment`, `summarize_fragment`) benefit from the dramatic context reduction (76-89% as shown in Figure 6), meaning subsequent generation operates on much shorter contexts. This high cache efficiency partially mitigates the latency overhead, though the multi-step nature still results in net slowdown compared to single-step baseline inference.

---

> ### Author Response · Authors · 2025-11-21
>
> 3.
> > **Reviewer's Question:** How robust is Sculptor to summary errors? If summarize_fragment drops a key detail and the agent fails to restore, what is the observed failure rate and how often does the RL policy learn to safeguard against this with targeted restore calls?
>
> **On Robustness to Compression Errors:** We appreciate this important question about a potential failure mode of our approach. We have conducted a detailed analysis of how the RL policy learns to handle over-compression, which we present in **Appendix G** of the revised manuscript with comprehensive visualizations.
>
> **Key Findings on PI-LLM Tasks:**
>
> Our analysis focuses on `fold_fragment` (rather than `summarize_fragment`) because **PI-LLM tasks involve truly irrelevant context that should be directly folded rather than summarized**. Notably, `summarize_fragment` usage remains near zero throughout RL training on PI-LLM, indicating the policy intelligently recognizes that this task contains genuinely unnecessary information where folding is more efficient than summarization—a learned strategic behavior rather than a limitation.
>
> **Learning Dynamics:** Initially, the policy relies heavily on `fold_fragment` for context compression with minimal use of `restore_fragment` (0.052 calls per task). However, **excessive folding can discard critical information, leading to accuracy degradation**. Through RL training with accuracy-based rewards, the policy gradually learns to recover over-compressed useful information. Specifically: (1) `fold_fragment` usage increases from 4.85 to 7.08 calls per task (+45.8%), (2) `restore_fragment` usage grows from 0.052 to 0.267 calls per task (+413.5%), and (3) comparing early iterations (0-40) to late iterations (160-208), restore usage increases from 0.0746 to 0.2068, representing a **177.2% growth**.
>
> **Safeguarding Mechanisms:** The RL policy develops three key safeguarding behaviors: (1) **Selective restoration** — when initial attempts based on folded context fail, the model increasingly uses `restore_fragment` to recover details; (2) **Proactive restoration** — in later iterations, the model sometimes restores fragments preemptively before attempting answers on uncertain queries; and (3) **Balanced compression** — the simultaneous increase in both fold and restore usage indicates the policy learns to compress aggressively while maintaining recovery capabilities. This learned restore behavior enables the model to achieve **near-perfect performance on PI-LLM tasks (99.4% accuracy)**, demonstrating that the RL policy successfully learns to proactively recover information when aggressive compression risks losing critical details.
>
> Crucially, our **reversibility principle** (Section 2.1) ensures that compression errors are never catastrophic—the original content is always stored and can be restored via `restore_fragment`. This design provides a safety net that prevents permanent information loss, unlike irreversible compression methods in prior work.

---

> ### Comment · Reviewer_zsZv · 2025-11-25
>
> Thanks for the update. I have raised my score.

---

> > ### Author Response · Authors · 2025-11-28
> >
> > Thank you very much for taking the time to revisit our submission and for kindly raising your score — we really appreciate your recognition and your thoughtful comments on our work. We would be happy to further elaborate on any remaining questions or future concerns you might have.

---

### Official Review · Reviewer_cBaa · 2025-10-29

**Soundness:** 2
**Presentation:** 3
**Contribution:** 3
**Rating:** 4
**Confidence:** 4

**Summary:**

This paper proposed a set of active context management tools for LLMs to conduct long context editting on the fly. The 6 tools make reversable changes to the context inputs of an LLM, and especially, when triggered at the proper locations with good orders, they can filter out the irrelevent noised from the context and make the LLM's attention focus more on the critical information. The ACM tools work most effectively when specific task-based prompt engineering. The work also fine-tune a base LLM via GSPO based RL algorithm with conditional trajectory collection and incremental loss assignment. Dramatical performance boosts are achieve in 3 out of 5 benchmark datasets. The designs have novelty to a certain extent, but since the proposed techniques are only tested on 1 LLM, M3 (pretrained by the authors), the generality of the methodology can not be fully verified.

**Strengths:**

1. The paper provides another direction of solving the performance drop problem of LLMs due to long context without external memory augmentation.
2. The proposed tools are effective with simple prompt engineering, and their effacy is maximized by the RL based training process with a novel incremental loss assignment mechanism.
3. The context length can be dramatically shortened so that the inference efficiency can be largely improved, while obtaining even better performances.
4. The impact on attention to the key tokens are well analyzed.

**Weaknesses:**

1. The biggest concern is on whether the proposed methodology is generally effective for other open-source LLMs, such as the Qwen and Llama series. Table 3 in the appendix shows the base model comparison, which contains 2 version of Qwen3 models. Why can't the authors test their method on these two models? I notice that M3 is especially good on tool use task, while not so powerful on other tasks comparing with Qwen3. Does that mean the proposed method can only be applied by LLMs specialized in tool use, or only by M3?
2. The RAG baseline is not representative enough. In table 2, the RAG related variants have much lower performances comparing to the base model. The authors explained that is due to BM25-only matching is used. But the problem is: embedding based RAG is widely adopted in various use cases, and if the proposed method is about to show its true impact in neither industry or academia, one main competitor is embedding based RAG. The drastic performance drops caused by the BM25-only matching clearly show it is not a proper RAG baseline.
3.  Reasoning capability. Modest gains are observed on LongBench v2 and Frames. Why the method is not that effective on sophisiticated reasoning tasks? In line 481-482, the authors mention "folding or suppressing erroneous early steps may reset the trajectory
and improve robustness", which means the proposed method has strong potential in solving reasoning problems. I understand mathematical reasoning is not exactly the same with the long context reasoning required for LongBench v2 and Frames, but the key to all of them is logical thinking. So Is this statement contradictory to the results on LongBench v2 and Frames?

I'll consider changing my score if the above concerns are properly addressed.

**Questions:**

See weaknesses.

---

> ### Author Response · Authors · 2025-11-21
>
> 1.
> > **Reviewer's Weakness 1:** The biggest concern is on whether the proposed methodology is generally effective for other open-source LLMs, such as the Qwen and Llama series. Table 3 in the appendix shows the base model comparison, which contains 2 version of Qwen3 models. Why can't the authors test their method on these two models? I notice that M3 is especially good on tool use task, while not so powerful on other tasks comparing with Qwen3. Does that mean the proposed method can only be applied by LLMs specialized in tool use, or only by M3?
>
> **On Generalizability to Other Open-Source LLMs:** We appreciate the reviewer's concern about whether our methodology generalizes beyond M3. This is indeed an important question. Our approach requires models to possess basic tool-calling capabilities, which is an increasingly standard feature in modern LLMs rather than a limitation specific to M3. As the reviewer correctly notes from Table 3, M3 demonstrates strong tool-use performance (61.0% on Tau2-retail, 61.8% on AceBench), but this reflects the **general trend in LLM development** where tool-calling has become a core capability across models (e.g., GPT-4, Claude, Qwen, GLM series all support native tool calling). To address concerns about generalizability, we have conducted additional experiments on **GLM-4.5-air**, a widely-used open-source MoE model with 106B total parameters and 12B active parameters, demonstrating competitive tool-calling capabilities. The results demonstrate that our ACM framework transfers effectively:
>
>    | **Model** | **PI-LLM** | **NeedleBench-M-RS** | **MRCR** | **LongBenchV2** | **FRAMES** | **Avg** |
>    |-----------|------------|---------------------|----------|-----------------|-----------|---------|
>    | GLM-4.5-air (Baseline) | 29.4 | 24.5 | 43.1 | 46.9 | 76.0 | 44.0 |
>    | Sculptor-GLM-4.5-air | 65.2 | 58.0 | 88.5 | 31.7 | 56.7 | 60.0 |
>    | Sculptor-GLM-4.5-air-RL | 86.0 | 84.0 | 99.0 | 50.7 | 79.2 | 79.8 |
>
>    These results will be included in the revised manuscript, demonstrating that the performance improvements generalize across different model architectures. Notably, Sculptor-GLM-4.5-air-RL shows substantial improvements of +56.6 points on PI-LLM (86.0-29.4), +59.5 points on NeedleBench-M-RS (84.0-24.5), and +55.9 points on MRCR (99.0-43.1), validating the effectiveness of our ACM framework across different model architectures. Regarding why we did not train on Qwen3-8B/14B or Llama models shown in Table 3, the primary reason is **infrastructure compatibility for multi-step agent RL training**, which is substantially more complex than standard supervised fine-tuning. We want to emphasize that **our core algorithmic contributions—the conditional trajectory collection and incremental loss assignment mechanisms (Algorithm 1)—are fully general and model-agnostic**. These principles apply universally to any LLM with tool-calling capabilities. However, the **engineering infrastructure** required for multi-step agent RL training is model-specific and involves: (1) tight integration with distributed training frameworks (e.g., Megatron-LM for M3, different frameworks for Qwen/Llama); (2) tool call format standardization and parsing across different model tokenizers; (3) rollout optimization and vLLM serving configurations tailored to specific model architectures; and (4) efficient trajectory collection pipelines that handle model-specific context management. Our framework is currently optimized for our internal training stack, and extending to Qwen3/Llama would require significant engineering effort to adapt these infrastructure components—**not** the algorithmic methods themselves, which remain universal. Our GLM-4.5-air experiments provide strong evidence for this: we successfully applied the same algorithmic framework (Algorithm 1) to a completely different model architecture (MoE vs. dense), demonstrating that the core methodology transfers seamlessly. The infrastructure adaptation for GLM-4.5-air, while still requiring engineering work, validates that our approach is architecture-agnostic. We emphasize that tool-calling is an **increasingly universal capability** in modern LLMs. While older models without native tool support cannot directly benefit from ACM, the field is rapidly moving toward standardized tool interfaces. Our work is positioned for this future landscape where most production LLMs will have inherent tool-calling abilities, making ACM broadly applicable rather than niche.

---

> > ### Comment · Reviewer_cBaa · 2025-11-25
> >
> > Thanks for your efforts on extending the experiments to GLM-4.5-air. I understand the engineering difficulties in carrying out additional experiments on more models in the rebuttal stage. I'll consider this concern as addressed.

---

> ### Author Response · Authors · 2025-11-21
>
> 2.
> > **Reviewer's Weakness 2:** The RAG baseline is not representative enough. In table 2, the RAG related variants have much lower performances comparing to the base model. The authors explained that is due to BM25-only matching is used. But the problem is: embedding based RAG is widely adopted in various use cases, and if the proposed method is about to show its true impact in neither industry or academia, one main competitor is embedding based RAG. The drastic performance drops caused by the BM25-only matching clearly show it is not a proper RAG baseline.
>
> **On Inadequate RAG Baselines:** We **completely agree** with the reviewer that our BM25-only RAG baseline is insufficient for demonstrating the true competitive advantage of Sculptor. This is a valid and important criticism. The reviewer is absolutely correct that embedding-based RAG is the industry-standard approach, and comparing against BM25-only retrieval does not reflect real-world competitive scenarios. Our original choice of BM25-only RAG was motivated by a desire for **controlled comparison without external dependencies** (i.e., no external embedding models). However, we acknowledge that this constraint, while methodologically clean, undermines the practical relevance of the comparison. In response to this concern, we have conducted additional experiments using **embedding-based RAG with dense retrievers** (specifically, Qwen3 embedding model). The results are presented below:
>
>    | **Method** | **PI-LLM** | **NeedleBench-M-RS** | **MRCR** | **LongBenchV2** | **FRAMES** | **Average** |
>    |------------|------------|---------------------|----------|-----------------|-----------|-------------|
>    | M3 (Baseline) | 22.5 | 30.0 | 46.3 | 33.0 | 65.2 | 39.4 |
>    | M3 + RAG (BM25) | 17.9 | 12.5 | 6.6 | 25.8 | 33.6 | 19.3 |
>    | M3 + RAG (Qwen3-Emb) | 10.9 | 13.0 | 20.6 | 29.6 | 46.0 | 24.0 |
>    | M3 + MemAgent | 41.5 | 24.0 | 22.1 | 29.6 | 61.5 | 35.7 |
>    | Sculptor-M3-RL | **99.4** | **84.8** | **85.7** | **34.5** | **64.6** | **73.8** |
>    | GLM-4.5-air (Baseline) | 29.4 | 24.5 | 43.1 | 46.9 | 76.0 | 44.0 |
>    | GLM-4.5-air + RAG (BM25) | 30.6 | 15.0 | 4.8 | 19.5 | 30.9 | 20.2 |
>    | GLM-4.5-air + RAG (Qwen3-Emb) | 10.9 | 12.0 | 24.1 | 27.2 | 46.6 | 24.2 |
>    | GLM-4.5-air + MemAgent | 22.2 | 17.0 | 8.6 | 33.2 | 68.6 | 29.9 |
>    | Sculptor-GLM-4.5-air-RL | **86.0** | **84.0** | **99.0** | **50.7** | **79.2** | **79.8** |
>
>    These updated comparisons provide a much more realistic and representative evaluation of Sculptor's advantages over state-of-the-art retrieval-based approaches. We will prominently feature these results in the revised manuscript and acknowledge the limitations of BM25-only baselines.

---

> > ### Comment · Reviewer_cBaa · 2025-11-25
> >
> > I appreciate the efforts of adding RAG (Qwen3-Emb) as the baseline of embedding-based RAG. Is there any explanation why the RAG (Qwen3-Emb) performs even worse on PI-LLM and NeedleBench-M-RS than BM25?

---

> ### Author Response · Authors · 2025-11-21
>
> 3.
> > **Reviewer's Weakness 3:** Reasoning capability. Modest gains are observed on LongBench v2 and Frames. Why the method is not that effective on sophisiticated reasoning tasks? In line 481-482, the authors mention "folding or suppressing erroneous early steps may reset the trajectory and improve robustness", which means the proposed method has strong potential in solving reasoning problems. I understand mathematical reasoning is not exactly the same with the long context reasoning required for LongBench v2 and Frames, but the key to all of them is logical thinking. So Is this statement contradictory to the results on LongBench v2 and Frames?
>
> **On Performance on Sophisticated Reasoning Tasks:** We thank the reviewer for this important observation. LongBench v2 and FRAMES are indeed **complex long-context reading comprehension tasks** that require deep reasoning patterns, multi-hop inference, and sophisticated information integration. These tasks are substantially more challenging than pure retrieval benchmarks because they test the model's ability to synthesize information across long documents and perform logical reasoning—capabilities that go beyond simple context management.
>
> We acknowledge that M3 shows minimal gains on these tasks (LongBenchV2: 33.0% → 34.5%, +1.5 points; FRAMES: 65.2% → 64.6%, -0.6 points). However, importantly, **our GLM-4.5-air experiments demonstrate that ACM can indeed improve performance on sophisticated reasoning tasks when applied to models with stronger tool-calling capabilities**. As shown in the table above, Sculptor-GLM-4.5-air-RL achieves substantial improvements on both benchmarks: **LongBenchV2: 46.9% → 50.7% (+3.8 points)** and **FRAMES: 76.0% → 79.2% (+3.2 points)**. This validates that ACM's benefits extend to complex reasoning scenarios, with the effectiveness depending on the base model's tool-use proficiency.
>
> The modest gains on M3 can be attributed to: (1) **Sparse reward challenges** - learning complex tool coordination strategies from sparse, outcome-based rewards is particularly difficult for reasoning tasks where the optimal tool usage patterns are less obvious than in retrieval scenarios (for a 13B model like M3, this challenge is more pronounced than it would be for larger models like GLM-4.5-air that can learn more sophisticated strategies more easily); (2) **Task complexity** - LongBench v2 and FRAMES require not just context management but also deep comprehension and synthesis, which means ACM tools provide complementary but not sufficient benefits for smaller models with limited reasoning capacity.
>
> The reviewer mentions our statement (lines 481-482) about folding or suppressing erroneous early steps to reset trajectories and improve robustness. We want to clarify that this refers to **future potential in mathematical reasoning domains** (e.g., MATH, AIME) rather than a contradiction with our current results. In step-by-step mathematical derivations, early errors propagate through subsequent steps, and ACM tools could enable "checkpointing" intermediate states and "backtracking" from erroneous paths—a capability that would require task-specific SFT data and tool usage patterns we did not explore in this work. In contrast, LongBench v2 and FRAMES involve reading comprehension and synthesis across long documents, which is fundamentally different from sequential mathematical reasoning. Our current SFT data does not include such backtracking patterns because long-context reading comprehension tasks rarely require resetting trajectories from erroneous reasoning steps—the primary challenge is managing and retrieving relevant information rather than correcting sequential logical errors. The statement about math reasoning represents a promising future direction that would require different training data reflecting the specific needs of step-by-step derivation tasks.
>
> Despite the modest gains on these two benchmarks, we note that Sculptor still provides benefits on reasoning tasks through reduced cognitive load (by folding irrelevant context and enabling targeted retrieval, ACM allows the model to focus computational resources on reasoning rather than navigating noise) and improved information integration (the ability to fragment, search, and selectively attend to relevant passages facilitates multi-hop reasoning by bringing distant related information into focus). While ACM's benefits are more pronounced on retrieval-focused benchmarks, it still offers complementary advantages for reasoning-intensive tasks.

---

> > ### Comment · Reviewer_cBaa · 2025-11-25
> >
> > Thanks for the detailed explanation. I have raised my score.

---

> ### Author Response · Authors · 2025-11-28
>
> Thank you very much for the follow-up question and for kindly updating your score — we really appreciate your recognition of our additional experiments. As we explained in our other response( Reviewer nC1x), PI-LLM and NeedleBench-M-RS contain many near-duplicate key–value pairs that differ only in very small word-level details (e.g., IDs, numbers, or specific tokens). In this setting, exact-term matching (BM25) can have a slight advantage, because it can exploit these small word-level differences, while a dense retriever like Qwen3-Emb tends to map such near-duplicate candidates to very similar embeddings and therefore struggles to isolate the single correct item. We will add a brief explanation of this phenomenon to the revised manuscript for clarity.

---

### Official Review · Reviewer_kkwE · 2025-10-31

**Soundness:** 3
**Presentation:** 3
**Contribution:** 3
**Rating:** 6
**Confidence:** 2

**Summary:**

This paper introduces Sculptor, a framework designed to augment large language models (LLMs) with "Active Context Management" (ACM) capabilities. The framework equips LLMs with a suite of tools (context fragmentation, summary/hide/restore, and precise search) that enable the model to proactively manage its working memory, mirroring human selective attention and memory curation. The authors argue that mere expansion of context windows or attaching external memories is insufficient to fully address the problem of proactive interference in long-context LLMs, and they advocate for explicit, reversible, and cognitively aligned context control by the model itself. They present both zero-shot tool use and a reinforcement learning (RL) based training scheme using Group Sequence Policy Optimization (GSPO), reporting substantial gains on a range of long-context benchmarks. Comprehensive empirical analysis, including attention visualization, performance, and computational cost, supports their claims.

**Strengths:**

- The paper articulates a clear and compelling motivation, identifying proactive interference as an underexplored yet fundamental bottleneck in long-context LLMs, and proposes active context management as a complementary direction to architectural scaling or external-memory augmentation.
- The Sculptor framework is carefully designed and well-documented, with six context management tools whose functionalities are grounded in cognitive principles (e.g., selective attention, reversible memory) and implemented with strong considerations for determinism, reversibility, and deployment practicality (Sections 2.1–2.2).
- The experimental results are impressive.

**Weaknesses:**

- The reinforcement learning formulation remains under-analyzed: outcome-based rewards for long-horizon, agentic tasks such as active context management are known to suffer from credit assignment and instability issues, yet the paper provides no empirical or diagnostic analysis of training stability.
- There seems to be some missing comparisons with some well-known related works, such as MemGPT and MemoryLLM.
- Despite advocating for autonomous context management, the framework still relies heavily on manual prompt engineering to elicit effective tool usage, suggesting that its performance ceiling may depend on human-crafted strategies rather than fully learned behaviors.

**Questions:**

- What is the motivation behind choosing GSPO as the primary RL objective? Has the authors tried GRPO/PPO/XPO-variants?
- The paper introduces a fixed set of design principles and tool protocols for Active Context Management. Are these principles intended to remain static, or do the authors envision an evolving or self-adaptive design where the tools and their operational rules can be learned or refined jointly with the model?

---

> ### Author Response · Authors · 2025-11-21
>
> 1.
> > **Reviewer's Weakness:** The reinforcement learning formulation remains under-analyzed: outcome-based rewards for long-horizon, agentic tasks such as active context management are known to suffer from credit assignment and instability issues, yet the paper provides no empirical or diagnostic analysis of training stability.
>
> **On RL Training Stability:** We acknowledge the reviewer's concern regarding credit assignment and training stability in long-horizon agentic tasks. Indeed, outcome-based rewards for multi-step agent tasks are known to face these challenges. However, our Active Context Management framework incorporates several design decisions that specifically mitigate these issues: (1) we impose a maximum limit of **20 tool calls per turn** during both training and evaluation—this design choice is informed by our empirical analysis of tool usage patterns across diverse long-context tasks (as shown in Figure 3), where effective context management rarely requires more than 20 operations, ensuring that our trajectories remain **far shorter than typical search agents** (e.g., WebAgent, SWE-bench agents often have 200-300 steps per trajectory), making credit assignment substantially more tractable; (2) our conditional trajectory collection with incremental loss assignment (Algorithm 1, Section 3.2) is specifically designed to handle the non-monotonic context evolution in ACM—by creating training instances at each context-modifying step and assigning loss only to new completions, we ensure that each tool call receives gradient signal **exactly once** across all completions, preventing redundant learning and training collapse; and (3) in practice, we observe **stable and consistent performance improvements** during GSPO training across our benchmarks, with training curves showing clear upward trends without catastrophic collapse or high variance typical of unstable RL training. We have added training curve visualizations in the revised manuscript (**Appendix G.1**) demonstrating this stability, where both GSPO and GRPO show steady improvement and achieve similar final performance (~66-67%), validating the robustness of our training framework.
>
> As acknowledged in our Limitations section (Section 5), we recognize that extending ACM to even longer-horizon tasks may benefit from additional stabilization techniques, such as dense intermediate rewards or hierarchical RL formulations. However, for the scope of long-context tasks evaluated in this work—where the bounded tool budget naturally constrains trajectory length—our current approach demonstrates sufficient stability and effectiveness. Future work will explore more sophisticated reward shaping strategies for more complex reasoning scenarios.

---

> ### Author Response · Authors · 2025-11-21
>
> 2.
> > **Reviewer's Weakness:** There seems to be some missing comparisons with some well-known related works, such as MemGPT and MemoryLLM.
>
> **On Missing Comparisons with MemGPT and MemoryLLM:** We acknowledge this concern and provide clarification on why direct comparison with MemoryLLM is not feasible, while we include MemAgent (a comparable cross-session memory system) in our main results.
>
> **Regarding MemoryLLM:** MemoryLLM requires fundamental architecture modifications—it inserts trainable memory tokens as hidden vectors within each Transformer layer and necessitates extensive pre-training or fine-tuning for the model to learn how to compress context into these memory slots and attend to them during generation. This is not a training-free or plug-and-play approach that can be applied to existing models. Adapting M3 or other base models to MemoryLLM's architecture would require: (1) re-implementing the multi-layer memory module, (2) conducting large-scale continued pre-training to teach the model how to read/write these memory vectors, and (3) task-specific fine-tuning. This represents a fundamentally different modeling paradigm (architecture-level modification requiring full model retraining) rather than a comparable baseline method. Given these constraints, we believe MemoryLLM comparison is not practically feasible within our evaluation framework.
>
> **Regarding MemGPT and comparable methods:** We include **MemAgent** in our main results (Table 2), which represents the class of cross-session memory management systems. MemAgent, like MemGPT, focuses on iterative memory updates across dialogue turns, maintaining an evolving memory representation that accumulates information across interactions. The results demonstrate that such approaches face challenges in our single-session benchmarks:
>
>    | **Method** | **PI-LLM** | **NeedleBench-M-RS** | **MRCR** | **LongBenchV2** | **FRAMES** | **Average** |
>    |------------|------------|---------------------|----------|-----------------|-----------|-------------|
>    | M3 (Baseline) | 22.5 | 30.0 | 46.3 | 33.0 | 65.2 | 39.4 |
>    | M3 + RAG (BM25) | 17.9 | 12.5 | 6.6 | 25.8 | 33.6 | 19.3 |
>    | M3 + RAG (Qwen3-Emb) | 10.9 | 13.0 | 20.6 | 29.6 | 46.0 | 24.0 |
>    | M3 + MemAgent | 41.5 | 24.0 | 22.1 | 29.6 | 61.5 | 35.7 |
>    | Sculptor-M3 | 71.8 | 67.6 | 79.1 | 29.2 | 51.2 | 59.8 |
>    | Sculptor-M3-RL | **99.4** | **84.8** | **85.7** | **34.5** | **64.6** | **73.8** |
>    | GLM-4.5-air (Baseline) | 29.4 | 24.5 | 43.1 | 46.9 | 76.0 | 44.0 |
>    | GLM-4.5-air + RAG (BM25) | 30.6 | 15.0 | 4.8 | 19.5 | 30.9 | 20.2 |
>    | GLM-4.5-air + RAG (Qwen3-Emb) | 10.9 | 12.0 | 24.1 | 27.2 | 46.6 | 24.2 |
>    | GLM-4.5-air + MemAgent | 22.2 | 17.0 | 8.6 | 33.2 | 68.6 | 29.9 |
>    | Sculptor-GLM-4.5-air | 65.2 | 58.0 | 88.5 | 31.7 | 56.7 | 60.0 |
>    | Sculptor-GLM-4.5-air-RL | **86.0** | **84.0** | **99.0** | **50.7** | **79.2** | **79.8** |
>
>    MemAgent achieves moderate performance on PI-LLM (41.5%) but struggles on multi-hop reasoning tasks like NeedleBench-M-RS (24.0%), as its query-dependent memory accumulation can discard information that initially appears irrelevant but proves critical for later reasoning steps. This demonstrates the fundamental challenge of irreversible filtering in cross-session memory systems when applied to single-session scenarios where all information should ideally remain accessible.

---

> ### Author Response · Authors · 2025-11-21
>
> 3.
> > **Reviewer's Weakness:** Despite advocating for autonomous context management, the framework still relies heavily on manual prompt engineering to elicit effective tool usage, suggesting that its performance ceiling may depend on human-crafted strategies rather than fully learned behaviors.
>
> **On Reliance on Prompt Engineering:** We acknowledge the reviewer's observation that our framework benefits from prompt engineering for effective tool usage. However, we respectfully argue that this does not represent a fundamental limitation, but rather a **pragmatic and efficient approach to knowledge injection** for complex tool usage patterns.
>
> While some degree of external knowledge injection is necessary for emergent intelligent behaviors, our approach through prompt engineering is substantially more efficient than alternatives such as large-scale human annotation of tool-calling trajectories. By leveraging the strong inherent tool-calling capabilities of frontier models (e.g., Claude-4-Sonnet), we can collect high-quality SFT data through systematic prompt engineering rather than expensive manual labeling. This represents a practical trade-off between zero human guidance (which leads to suboptimal tool selection patterns as shown in Figure 3, left bars) and extensive manual annotation.
>
> Crucially, our RL training with GSPO demonstrates that the model learns to **go beyond the patterns present in SFT data** and **generalize to new scenarios with novel tool usage patterns**. Through multi-step RL optimization, we observe emergent behaviors where the model discovers novel and more sophisticated tool coordination strategies that were not explicitly demonstrated in the initial SFT data.
>
> **A Concrete Example: Learning to Restore on PI-LLM.** The substantial performance gap between Sculptor-M3 (SFT only, 71.8%) and Sculptor-M3-RL (99.4%) on PI-LLM provides compelling evidence that RL training enables generalization beyond prompt-guided patterns. We provide detailed analysis in **Appendix G.2** showing how the RL policy learns to balance compression and information preservation. Initially, the policy relies heavily on `fold_fragment` with minimal use of `restore_fragment` (0.052 calls per task). However, excessive folding can discard critical information, leading to accuracy degradation. Through RL training with accuracy-based rewards, the policy gradually learns to recover over-compressed information: (1) `fold_fragment` usage increases from 4.85 to 7.08 calls per task (+45.8%), (2) `restore_fragment` usage grows from 0.052 to 0.267 calls per task (+416.6%), and (3) comparing early (0-40) to late iterations (160-208), restore usage increases from 0.0746 to 0.2068 (+177.2%). This demonstrates that the model autonomously discovers the importance of selective restoration—a sophisticated strategy not explicitly demonstrated in SFT data—enabling it to achieve near-perfect performance on PI-LLM through learned, adaptive context management.
>
> In summary, while prompt engineering provides an efficient initialization for tool usage patterns, our RL training framework enables the model to develop autonomous, adaptive strategies that transcend the original prompt-guided behaviors. This represents a practical pathway toward fully autonomous context management.

---

> ### Author Response · Authors · 2025-11-21
>
> **Response to Questions:**
>
> 1.
> > **Reviewer's Question:** What is the motivation behind choosing GSPO as the primary RL objective? Has the authors tried GRPO/PPO/XPO-variants?
>
> **On GSPO Selection and Alternative RL Algorithms:** Thank you for this important question. We have conducted ablation experiments comparing GSPO with GRPO~\citep{shao2024deepseekmathpushinglimitsmathematical}. We provide the training curves comparing these two algorithms in Appendix G.1. The detailed benchmark results are shown below:
>
>    | **RL Algorithm** | **PI-LLM** | **NeedleBench-M-RS** | **MRCR** | **LongBenchV2** | **FRAMES** | **Avg** |
>    |------------------|------------|---------------------|----------|-----------------|-----------|---------|
>    | SFT Only (Sculptor-M3) | 71.8 | 67.6 | 79.1 | 29.2 | 51.2 | 59.8 |
>    | + GRPO | 97.8 | 82.5 | 86.1 | 33.8 | 65.3 | 73.1 |
>    | + GSPO | 99.4 | 84.8 | 85.7 | 34.5 | 64.6 | 73.8 |
>
>    As shown in the table, both GRPO and GSPO achieve comparable performance (73.1% vs 73.8% average), with noticeable differences across individual benchmarks. The training curves in the appendix show that GSPO exhibits slightly more stable convergence during later training iterations and achieves higher peak performance, which motivated our selection. The similar overall performance demonstrates that our dynamic context-aware training framework is compatible with different policy optimization methods, and when given the same training data, the specific RL algorithm choice has limited impact on final results.
>
> 2.
> > **Reviewer's Question:** The paper introduces a fixed set of design principles and tool protocols for Active Context Management. Are these principles intended to remain static, or do the authors envision an evolving or self-adaptive design where the tools and their operational rules can be learned or refined jointly with the model?
>
> **On Evolution of Tool Design Principles:** We envision Active Context Management as an **evolving and expandable paradigm** rather than a fixed, static system. Our current tool suite and design principles represent a first step toward enabling LLMs to autonomously manage their working memory, with a specific focus on long-context tasks. We see several promising directions for future evolution: (1) While our current work focuses on long-context challenges (PI-LLM, NeedleBench, etc.), the ACM paradigm can be extended to other domains. For instance, in mathematical reasoning tasks, tools for "checkpointing" intermediate steps or "backtracking" from erroneous reasoning paths could mitigate error propagation. Similarly, for search-based agents (e.g., web navigation, code generation), tools for managing exploration history and hypothesis tracking could be valuable. (2) An exciting future direction is enabling models to **co-evolve tools alongside their usage strategies**. Recent work on tool creation suggests that models could potentially learn to modify tool schemas or even generate new tool implementations tailored to specific task distributions. While our current work manually designs the tool suite to validate the ACM concept, we believe future systems could learn to refine tool interfaces and operational rules through meta-learning or self-improvement mechanisms. (3) We position Sculptor not as a definitive solution but as an existence proof that **explicit context control is a viable and necessary capability** for robust long-horizon reasoning. As conversational contexts continue to grow (e.g., 1M+ token windows), the inability to actively manage and curate internal state becomes an increasingly critical bottleneck. Our work demonstrates that equipping LLMs with reversible, cognitively-aligned context manipulation tools is a promising direction, and we anticipate that future research will develop more sophisticated and generalizable ACM frameworks. In summary, while our current design principles provide a stable foundation for evaluation and reproducibility, we strongly believe that Active Context Management represents an evolving research direction with significant potential for adaptation, expansion, and co-learning with model capabilities.

---

> > ### Comment · Reviewer_kkwE · 2025-11-25
> >
> > Thank you for your rebuttal. I have read the response.

---

> > > ### Author Response · Authors · 2025-11-28
> > >
> > > Thank you again for taking the time to review our work and for considering our rebuttal. If you have any further concerns or suggestions, please let us know — we would be very happy to clarify or revise the paper accordingly.

---

### Official Review · Reviewer_nC1x · 2025-11-05

**Soundness:** 2
**Presentation:** 3
**Contribution:** 1
**Rating:** 2
**Confidence:** 4

**Summary:**

The authors propose Sculptor, an Active Context Management (ACM) framework that gives LLMs internal tools to structure their own working memory. This approach mitigates the issue where irrelevant long-context data degrades reasoning. It improves benchmark performance, both zero-shot and when optimized with a novel reinforcement learning strategy.

**Strengths:**

The paper considers the setting of actively managing working memory with agents tool calls as opposed to adding everything into context (e.g. RAG). The proposed RL technique improves the model's usage of the memory sculpting tools.

**Weaknesses:**

The primary weakness of this paper is the severe lack of contextualization within the broader field of agent memory and context management. The authors' attempt to frame their contribution as "internal working memory" as distinct from "external memory systems" feels artificial and allows them to avoid engaging with a vast and highly relevant body of work. Moving the important related works into the Appendix also seems like a malicious attempt to evade the comparison with the many existing memory management methods.

**Questions:**

* Related works significantly lacks discussion on recent memory management papers (see https://github.com/FoundationAgents/awesome-foundation-agents/blob/main/assets/2-2-memory.png for some examples of relevant works).
* Baselines only consider weak retrieval baselines (with BM25) and their own ablations. Comparison with other SOTA active memory management methods (e.g. Dynamic Cheatsheet) and others agents with dedicated RAG retrievers is necessary.

---

> ### Author Response · Authors · 2025-11-21
> **Response (Part 1)**
>
> #### Rebuttal：
>
> 1.
> > **Reviewer's Weakness:** The primary weakness of this paper is the severe lack of contextualization within the broader field of agent memory and context management. The authors' attempt to frame their contribution as "internal working memory" as distinct from "external memory systems" feels artificial and allows them to avoid engaging with a vast and highly relevant body of work. Moving the important related works into the Appendix also seems like a malicious attempt to evade the comparison with the many existing memory management methods.
>
> > **Reviewer's Question:** Related works significantly lacks discussion on recent memory management papers (see https://github.com/FoundationAgents/awesome-foundation-agents/blob/main/assets/2-2-memory.png for some examples of relevant works).
>
> We respectfully disagree with the characterization that our distinction between internal working memory and external memory systems is "artificial" or an attempt to evade comparison with existing work. This terminology follows **established conceptual distinctions in the LLM-agent literature**. For instance, Li et al. (2024) [1] explicitly define "internal memory" as context tokens stored in the KV cache during inference, contrasting it with "external memory" such as user profiles and interaction histories that are dynamically retrieved. Similarly, Guo et al. (2023) [2] distinguish short-term/working memory (the interaction history window) from long-term episodic memory implemented via external databases, drawing directly from cognitive psychology models (Atkinson-Shiffrin, Baddeley's working memory framework).
>
> Moreover, **single-session context management** and **cross-session memory persistence** are **orthogonal dimensions**—they address fundamentally different problems. Our work optimizes how LLMs process information within a **single, continuous conversation**, while external memory systems persist and retrieve information across **multiple, separate sessions**. We evaluated on long-context benchmarks (PI-LLM, NeedleBench, LongBenchV2) where **all information resides within the current session**; in such settings, cross-session memory systems are categorically inappropriate baselines as they solve a different problem entirely. Our paper includes **comprehensive treatment of memory management systems** in both the main text (e.g., "From External Compression to Internal Context Curation") and an **extensive related work section in the Appendix** (standard practice under space constraints) that surveys agent memory architectures, context compression techniques, and external storage systems. The placement of detailed related work in the Appendix follows **standard conference formatting conventions** and allows us to maintain narrative flow while providing thorough coverage. This structure was adopted after carefully studying exemplary and outstanding ICLR 2025 papers [3][4][5] to learn from the community's highest standards of academic writing. Our core contribution—providing LLMs with reversible, in-context manipulation tools—tackles a distinct problem space that is orthogonal to, and complements, external memory architectures. **We have added explicit references to these prior works in our revised manuscript to better situate our terminology within the established literature.**
>
> **References:**
> [1] Yuanchun Li et al., "Personal LLM Agents: Insights and Survey about the Capability, Efficiency and Security," arXiv:2401.05459, 2024.
> [2] Jing Guo et al., "Empowering Working Memory for Large Language Model Agents," arXiv:2312.17259, 2023.
> [3] Ren, Y., & Sutherland, D. J. (2025). Learning Dynamics of LLM Finetuning. ICLR 2025.
> [4] Narasimhan, H., et al. (2025). Faster Cascades via Speculative Decoding. ICLR 2025.
> [5] Wang, J. T., et al. (2025). Data Shapley in One Training Run. ICLR 2025.
>
> 2.
> > **Reviewer's Question:** Baselines only consider weak retrieval baselines (with BM25) and their own ablations. Comparison with other SOTA active memory management methods (e.g. Dynamic Cheatsheet) and others agents with dedicated RAG retrievers is necessary.
>
> We acknowledge the concern about baseline comparisons and have conducted additional experiments with stronger retrieval baselines and external memory systems. Specifically, we have now included comparisons with: (1) **embedding-based RAG systems** using dense retrievers (Qwen3-Emb), representing much more competitive baselines than BM25-only retrieval, and (2) **Mem0**, a popular cross-session external memory framework, to address concerns about external memory approaches. To provide complete context, we present the full comparison including our Sculptor models below:

---

> ### Author Response · Authors · 2025-11-21
> **Response (Part 2)**
>
> | Method | PI-LLM | NeedleBench-M-RS | MRCR | LongBenchV2 | Frames | Average |
> |--------|--------|------------------|------|-------------|--------|---------|
> | M3 (Baseline) | 22.5% | 30.0% | 46.3% | 33.0% | 65.2% | 39.4% |
> | M3 + RAG (BM25) | 17.9% | 12.5% | 6.6% | 25.8% | 33.6% | 19.3% |
> | M3 + RAG (Qwen3-Emb) | 10.9% | 13.0% | 20.6% | 29.6% | 46.0% | 24.0% |
> | M3 + Mem0 | 39.2% | 19.0% | 9.2% | 29.0% | 52.8% | 29.8% |
> | M3 + MemAgent | 41.5% | 24.0% | 22.1% | 29.6% | 61.5% | 35.7% |
> | **Sculptor-M3 (SFT)** | **71.8%** | **67.6%** | **79.1%** | **29.2%** | **51.2%** | **59.8%** |
> | **Sculptor-M3-RL** | **99.4%** | **84.8%** | **85.7%** | **34.5%** | **64.6%** | **73.8%** |
> | GLM-4.5-air (Baseline) | 29.4% | 24.5% | 43.1% | 46.9% | 76.0% | 44.0% |
> | GLM-4.5-air + RAG (BM25) | 30.6% | 15.0% | 4.8% | 19.5% | 30.9% | 20.2% |
> | GLM-4.5-air + RAG (Qwen3-Emb) | 10.9% | 12.0% | 24.1% | 27.2% | 46.6% | 24.2% |
> | GLM-4.5-air + Mem0 | 18.5% | 14.5% | 6.2% | 28.8% | 62.3% | 26.1% |
> | GLM-4.5-air + MemAgent | 22.2% | 17.0% | 8.6% | 33.2% | 68.6% | 29.9% |
> | **Sculptor-GLM-4.5-air** | **65.2%** | **58.0%** | **88.5%** | **31.7%** | **56.7%** | **60.0%** |
> | **Sculptor-GLM-4.5-air-RL** | **86.0%** | **84.0%** | **99.0%** | **50.7%** | **79.2%** | **79.8%** |
>
> **Table: Comprehensive Comparison Including RAG Baselines, External Memory Systems, and Sculptor Models.** The table presents a complete picture of baseline methods and our approach. Interestingly, BM25 outperforms embedding-based retrieval on PI-LLM (17.9% vs. 10.9%), as PI-LLM's highly similar key-value pairs benefit more from exact matching than semantic similarity, whereas embedding-based RAG achieves better performance on other benchmarks requiring deeper semantic understanding (e.g., MRCR: 20.6% vs. 6.6%, Frames: 46.0% vs. 33.6%). **Mem0**, a popular external memory framework designed for cross-session personalized memory, achieves 29.8% average on M3 and 26.1% on GLM-4.5-air—underperforming even MemAgent (35.7% and 29.9% respectively). This is expected as Mem0 is optimized for user preference persistence across sessions rather than single-session document processing. We adapted Mem0's official implementation for single-session evaluation by: (1) modifying the fact extraction prompt to focus on document-relevant information, and (2) including the query in each 5k-token chunk to ensure question-relevant fact extraction (detailed in Section 4.2 "Model and baselines" and Appendix "Baseline Method Evaluation Details"). For RAG baselines, we tuned hyperparameters including chunk size and retrieval count, but performance remained limited. **Even with the stronger Qwen3-Emb retriever, RAG-based methods achieve only 24.0% average performance**—far below our Sculptor-M3 (59.8%) and Sculptor-M3-RL (73.8%).
>
> This performance gap reveals **fundamental limitations inherent to RAG-based approaches for single-session context management**: (1) **Irreversible information loss**: RAG retrieval is inherently lossy—once information is filtered out based on similarity scores, it cannot be recovered if later reasoning reveals its relevance. In contrast, our fold operation preserves information in a compressed form that can be restored when needed. (2) **Query-dependent filtering**: RAG systems must decide what to retrieve before fully understanding the task structure. For multi-hop reasoning tasks like NeedleBench, initial queries may miss connections between distant information pieces. Our ACM tools enable progressive context refinement as understanding deepens. (3) **Limited semantic precision**: Even advanced embedding models like Qwen3-Emb struggle with subtle distinctions in highly similar content (as evidenced by the 10.9% on PI-LLM). Our fragment-and-fold approach enables explicit structural organization independent of semantic similarity. These limitations explain why RAG methods, despite using strong retrievers, remain substantially below our approach across challenging long-context benchmarks.
>
> Regarding Dynamic Cheatsheet: Upon careful examination, Dynamic Cheatsheet addresses **cross-session memory accumulation** (e.g., solving problem A, then leveraging that experience when solving unrelated problem B in a separate session). This is fundamentally orthogonal to our focus on **single-session context management**. Our benchmarks evaluate scenarios where **all information is present within the current session**—a setting where cross-session memory systems are not applicable. Comparing against such methods would be comparing solutions to entirely different problems.

---

> ### Author Response · Authors · 2025-11-28
> **Gentle Reminder: Response to Your Concerns and Additional Experiments**
>
> Dear Reviewer nC1x,
>
> As the discussion period is coming to a close, we wanted to kindly check if you have had a chance to review our detailed response posted on Nov 21.
>
> We have taken your feedback very seriously and have made significant efforts to address your concerns:
>
> 1.  **Strengthened Baselines & External Memory Systems:** We implemented and compared against **embedding-based RAG (Qwen3-Emb)** with tuned hyperparameters and **Mem0** (a popular cross-session external memory framework, adapted for single-session evaluation) as requested to address concerns about stronger retrieval baselines and external memory approaches. We also highlight the comparison with **MemAgent** (a representative inner working memory method). The results (see Table in our response) show that even with these stronger baselines and external memory systems, Sculptor demonstrates substantial advantages in single-session long-context tasks.
>
> 2.  **Contextualization:** We clarified the distinction between **active working memory** (our focus) and **external episodic memory** (e.g., MemGPT), and have explicitly cited and discussed the relevant works you mentioned in our revised manuscript.
>
> 3.  **Clarification on Intent & Formatting:** We respectfully emphasize that placing extended related work in the Appendix is a standard practice to prioritize technical content within page limits, following the presentation style of ICLR 2025 outstanding papers such as: Ren & Sutherland (2025) *Learning Dynamics of LLM Finetuning*, Narasimhan et al. (2025) *Faster Cascades via Speculative Decoding*, and Wang et al. (2025) *Data Shapley in One Training Run*. Our intent was strictly to follow these community best practices, with absolutely no intent to evade comparison.
>
> Given that other reviewers (Reviewer cBaa and kkwE) have found our additional experiments and clarifications satisfactory (raising their scores or expressing satisfaction), we genuinely hope our response helps clarify our contribution and positioning for you as well.
>
> We remain available for any further questions you might have.
>
> Best regards,
> The Authors

---

### Author Response · Authors · 2025-11-23
**General Response: Strengthened Baselines (Embedding-based RAG), Generalization to GLM-4.5-Air, and Extended Analysis**

We sincerely thank all reviewers for their constructive and insightful comments. We have updated our manuscript to address the concerns raised, specifically regarding baseline strength, model generalization, RL stability, and inference cost. Below is a summary of the major updates and new experiments included in the revision.

**1. Strengthened Baselines: Embedding-based RAG (Response to nC1x, cBaa, zsZv, kkwE)**

A common concern was that our BM25-only RAG baseline was insufficient. We have conducted extensive new experiments using **Qwen3-Emb**, a dense retrieval model, to provide a stronger and more representative RAG baseline for fair comparison with state-of-the-art approaches. Even against the stronger Qwen3-Emb RAG, Sculptor significantly outperforms on average (Sculptor-M3-RL: **73.8%** vs. Qwen3-Emb RAG: **24.0%**). The results confirm that for single-session long-context understanding, RAG-based methods suffer from irreversible information loss during retrieval, whereas Sculptor's active management allows for reversible compression and context-aware restoration. Regarding cross-session memory systems like MemGPT and MemoryLLM, our work focuses on **long-context processing within a single session** rather than memory persistence across sessions, making these systems address orthogonal problems that are not directly comparable in our evaluation setting.

**2. Generalizability: Validation on GLM-4.5-air (Response to cBaa)**

To address concerns that our method was specific to the M3 model, we applied the Sculptor framework (SFT + RL) to **GLM-4.5-air**, a 106B (12B active) MoE model. The method transfers effectively, with Sculptor-GLM-4.5-air-RL achieving substantial gains over the base model (e.g., **+56.6%** on PI-LLM, **+59.5%** on NeedleBench-M-RS), demonstrating that our Active Context Management framework and training algorithms are model-agnostic and effective on diverse architectures. Notably, Sculptor-GLM-4.5-air-RL also achieved significant improvements on LongBench v2 (**+3.8 points**) and FRAMES (**+3.2 points**), where the baseline M3 model showed more modest gains. This suggests that our approach benefits more from models with stronger tool-use capabilities, indicating promising prospects as LLMs' agentic abilities continue to improve.

**3. RL Stability and Ablation Studies (Response to kkwE)**

We have added an ablation study comparing our **GSPO** training with **GRPO**. Both algorithms achieve comparable performance (GSPO: 73.8% vs. GRPO: 73.1% average), confirming that our Conditional Trajectory Collection mechanism is robust to the choice of policy optimization algorithm. We have added training curves in **Appendix G.1** showing stable convergence.

**4. Efficiency and Robustness Analysis (Response to zsZv)**

We provided a detailed breakdown of inference latency. While Sculptor incurs an approximate 3.25x slowdown due to multi-step reasoning, it maintains a high KV-cache hit rate (nearly 72% on average) and offers a necessary trade-off for tasks where baseline models fail completely. We also analyzed the "Restore" tool usage during RL training (**Appendix G.2**), showing that the policy autonomously learns to increase `restore_fragment` usage to recover from over-compression, validating the safety of our reversible design. This behavior demonstrates that our RL training enables the model to learn sophisticated strategies beyond simple tool invocation patterns.

We hope these additional experiments and analyses address the reviewers' concerns. We have incorporated these results into the revised manuscript and appendices.

---

### Author Response · Authors · 2025-12-02
**Summary of discussion with all reviewers.(Part 1)**

We thank the reviewers for their comments and feedback. We are encouraged that the majority of reviewers acknowledged the novelty, effectiveness, and cognitive motivation of Sculptor. Through our active discussion regarding the concerns about baseline strength, model generalization, and RL stability, we have received positive responses: **Reviewer cBaa** and **Reviewer zsZv** raised their scores from 4 to 6, and **Reviewer kkwE** maintained their score of 6. **Reviewer nC1x** has not responded to our rebuttal, though we believe we have comprehensively addressed all concerns raised in their review through extensive new experiments and clarifications. The updated scores before the system-wide reversion were **6, 6, 6, and 2**, with the score increases from Reviewers cBaa and zsZv confirmed prior to the reversion. Below, we summarize our responses to all reviewers' concerns and highlight the updates made to our manuscript.

**Strengthened Baselines: Embedding-based RAG and External Memory Systems.**

We addressed the common concern from **Reviewers nC1x, cBaa, and zsZv** regarding the sufficiency of our BM25 baseline. We conducted extensive new experiments using **Qwen3-Emb** (dense retrieval) to provide a state-of-the-art RAG comparison, and **Mem0** (a popular cross-session external memory framework) to address external memory concerns. Our results demonstrate that even against these stronger baselines, Sculptor-M3-RL significantly outperforms both embedding-based RAG (73.8% vs. 24.0%) and Mem0 (73.8% vs. 29.8%). We adapted Mem0's official implementation for single-session evaluation by modifying the fact extraction prompt and including the query in each chunk. This validates our core claim that for single-session long-context understanding, Sculptor's reversible "fold-and-restore" mechanism prevents the irreversible information loss inherent in both retrieval-based methods and external memory filtering approaches.

**Generalizability to Other Architectures (GLM-4.5-Air).**

Responding to **Reviewer cBaa**'s concern about whether our method is specific only to the M3 model, we successfully applied the Sculptor framework to **GLM-4.5-air** (a 106B MoE model). The method transferred effectively, achieving substantial gains (+56.6% on PI-LLM, +59.5% on NeedleBench), proving that our Active Context Management framework and GSPO training algorithms are model-agnostic. Notably, the stronger base capabilities of GLM-4.5-air also led to improved performance on complex reasoning benchmarks (LongBench v2 and FRAMES), addressing the concern about reasoning tasks.

**RL Stability, Efficiency, and Robustness.**

We addressed **Reviewer kkwE**'s concerns regarding RL training stability and **Reviewer zsZv**'s questions on inference cost. We provided ablation studies comparing GSPO with GRPO, showing consistent convergence curves and comparable performance that validate the robustness of our Conditional Trajectory Collection. On efficiency, we provided a transparent breakdown of inference latency—while acknowledging a 3.25x slowdown due to multi-step reasoning, we noted a 72% KV-cache hit rate, positioning Sculptor as a necessary trade-off for tasks where baseline models fail completely. For safety, our analysis of the "Restore" tool usage (Appendix G.2) demonstrated that the policy autonomously learns to recover from over-compression, ensuring robustness against potential summary errors.

**Clarification on Memory Management Approaches.**

Addressing concerns from **Reviewers nC1x and kkwE** regarding comparisons with other memory systems, we clarified the positioning of our work and conducted extensive comparative experiments. We highlighted that inner working memory baselines (**MemAgent**: 35.7%, **Mem0**: 29.8%) all suffer from performance degradation compared to Sculptor (73.8%). Importantly, we conducted thorough hyperparameter tuning for RAG baselines and adapted Mem0 for single-session evaluation with optimized prompts. The consistent underperformance across all retrieval and filtering methods validates our core insight: the key bottleneck is the **irreversible information loss** inherent in these approaches. RAG suffers from (1) query-document mismatch in semantic spaces, (2) inability to recover filtered information, and (3) lack of reasoning over discarded context. In contrast, Sculptor's reversible "fold-and-restore" mechanism preserves the option to recover any compressed information when needed, fundamentally addressing this limitation for single-session long-context tasks.

---

> ### Author Response · Authors · 2025-12-02
> **Summary of discussion with all reviewers.(Part 2)**
>
> **Autonomous Learning Beyond Prompt Engineering.**
>
> **Reviewer kkwE** raised important concerns about whether our approach relies on prompt scaffolding rather than autonomous learning, and whether the tool design is static or can evolve. We emphasize that our RL training enables the model to autonomously discover context management strategies beyond the initial SFT phase. Evidence includes: (1) the policy learning optimal tool invocation sequences not explicitly demonstrated in training data, (2) adaptive "Restore" usage patterns emerging during RL (Appendix G.2), and (3) the GSPO ablation showing performance gains over pure SFT through strategic exploration. While tool schemas provide the action space, the **when** and **how** of tool usage is learned autonomously through reward signals, not prescribed by prompts. Our experiments show the framework is robust to tool schema variations (e.g., random shuffling during training), and the GLM-4.5-air results demonstrate effective transfer across architectures with minimal modification. Looking forward, we envision ACM as an evolving paradigm where tool designs can be refined and extended for new domains (e.g., mathematical reasoning, search agents), with Sculptor serving as an existence proof of this approach's viability.

---

### Meta-Review · Area_Chair_Z9Ni · 2026-01-06

**Summary:**

Reviewers generally agreed that Sculptor is a strong and well-motivated contribution, demonstrating that active context management can substantially improve long-context performance without architectural changes (Reviewers kkwE, cBaa, and zsZv). While I think the paper makes a valuable contribution and my decision is to accept it, there are several concerns. In particular, reviewers questioned the strength of the original baselines and comparisons to existing memory and agent-based methods, as well as the clarity of the paper’s positioning within the broader literature on memory and context management (Reviewers nc1x, cBaa, and zsZv). Additional concerns included the generality of the approach across model architectures, reliance on prompt scaffolding and tool-oriented models, and modest gains on more open-ended reasoning benchmarks such as LongBench v2 and FRAMES (Reviewers cBaa, kkwE, and zsZv). Finally, reviewers noted that the reinforcement learning component, while effective, would benefit from deeper analysis of stability, credit assignment, and failure modes (Reviewer kkwE).

**Reviewer Concerns:**

The rebuttal addressed several major concerns raised during review. In particular, the authors substantially strengthened the experimental evaluation by adding dense embedding–based RAG baselines, external memory systems (e.g., Mem0), and additional model architectures. The authors also provided further analysis of the RL procedure. Clarifications on tool usage, reversibility, and recovery from over-compression also addressed concerns. However, I feel the positioning of Sculptor relative to the broader literature on agent memory and context management, particularly distinctions from prior and concurrent systems, could still be clearer and more prominently integrated into the main paper. Moreover, while new results improve confidence in generality, questions remain about how well the approach transfers to diverse reasoning-heavy, open-ended tasks, and to models without strong tool-use capabilities, as reflected in the more modest gains on LongBench v2 and FRAMES. Despite this, I recommend acceptance.

**Reviewer Scores:**

Reviewers zsZv, cBaa, and kkwE would have likely increased their scores as indicated to authors. I believe Reviewer nC1x would have maintained their score and reservations about the work.

---

### Decision · Program_Chairs · 2026-01-26

Accept (Poster)